# A *Mycobacterium tuberculosis* surface protein recruits ubiquitin to trigger host xenophagy

Qiyao Chai[1,2], Xudong Wang[1,3], Lihua Qiang[1,2], Yong Zhang[1,2], Pupu Ge[1,2], Zhe Lu[1,2], Yanzhao Zhong[1,2], Bingxi Li[1], Jing Wang[1], Lingqiang Zhang[4], Dawang Zhou[5], Wei Li[6], Wenzhu Dong[7], Yu Pang[7], George Fu Gao[1,2] & Cui Hua Liu[1,2]

Ubiquitin-mediated xenophagy, a type of selective autophagy, plays crucial roles in host defense against intracellular pathogens including *Mycobacterium tuberculosis* (Mtb). However, the exact mechanism by which host ubiquitin targets invaded microbes to trigger xenophagy remains obscure. Here we show that ubiquitin could recognize Mtb surface protein Rv1468c, a previously unidentified ubiquitin-binding protein containing a eukaryotic-like ubiquitin-associated (UBA) domain. The UBA-mediated direct binding of ubiquitin to, but not E3 ubiquitin ligases-mediated ubiquitination of, Rv1468c recruits autophagy receptor p62 to deliver mycobacteria into LC3-associated autophagosomes. Disruption of Rv1468c-ubiquitin interaction attenuates xenophagic clearance of Mtb in macrophages, and increases bacterial loads in mice with elevated inflammatory responses. Together, our findings reveal a unique mechanism of host xenophagy triggered by direct binding of ubiquitin to the pathogen surface protein, and indicate a diplomatic strategy adopted by Mtb to benefit its persistent intracellular infection through controlling intracellular bacterial loads and restricting host inflammatory responses.

[1] Chinese Academy of Sciences Key Laboratory of Pathogenic Microbiology and Immunology, Institute of Microbiology, Chinese Academy of Sciences, Beijing 100101, China. [2] Savaid Medical School, University of Chinese Academy of Sciences, Beijing 101408, China. [3] College of Life Sciences, University of Chinese Academy of Sciences, Beijing 100049, China. [4] State Key Laboratory of Proteomics, Beijing Proteome Research Center, National Center of Protein Sciences (Beijing), Beijing Institute of Lifeomics, Beijing 100850, China. [5] State Key Laboratory of Cellular Stress Biology, Innovation Center for Cell Signaling Network, School of Life Sciences, Xiamen University, Xiamen, Fujian 361102, China. [6] State Key Laboratory of Stem Cell and Reproductive Biology, Institute of Zoology, Chinese Academy of Sciences, Beijing 100101, China. [7] Beijing Tuberculosis and Thoracic Tumor Research Institute, Beijing Chest Hospital, Capital Medical University, Beijing 101149, China. Correspondence and requests for materials should be addressed to C.H.L. (email: liucuihua@im.ac.cn)

biquitin (Ub) targeting to intracellular bacteria plays a fundamental role in selective autophagy called xenophagy, a crucial innate immune mechanism in mammalian cells against intracellular pathogens[1]. During the infection, various types of Ub chains, such as K63 and K48 Ub chains, have been demonstrated to form Ub coat surrounding the pathogens for recognition by a range of autophagy receptors, including p62 (SQSTM1), NBR1, and NDP52, which couple the Ub-decorated substrates and LC3, an autophagosomal membrane-associated protein, to trap bacteria into autophagosomes[2–5]. But how the host Ub attaches to the pathogens for triggering xenophagy is still a mysterious process. Previous studies showed that several ubiquitin-ligating (E3) enzymes such as RNF166, LRSAM1, Parkin, and Smurf1 play important roles in directing Ub-coated intracellular bacterial pathogens or the bacteria-containing vacuoles to autophagosomes[3,4,6,7]. However, the specific substrate proteins targeted by those E3 Ub ligases for ubiquitination remain unclear[1,8,9]. *Mycobacterium tuberculosis* (Mtb) is an ancient successful intracellular pathogen for causing tuberculosis (TB). Data from previous studies showed that the colocalization of Ub with Mtb was not completely absent in macrophages deficient in Parkin and/or Smurf1[3,4], hinting that additional mechanisms involved in mediating the targeting of Ub to mycobacteria.

Apart from being covalently attached to the lysine residues on protein substrates through E3 Ub ligases-mediated ubiquitination, Ub could also hydrophobically interact with Ub-binding proteins (UBPs, known as Ub receptors) that contain the Ub-binding domains (UBDs), which process is independent of E3 Ub ligases[10]. We previously found that the mycobacterial effector protein PtpA contains an Ub-interacting motif-like (UIML) region for host Ub binding and innate immune suppression[11], which discovery prompted us to wonder whether there are certain Mtb surface proteins that could be directly targeted by host Ub for triggering xenophagy-mediated bacterial clearance. Such information could be important for developing novel Mtb-host interface-based anti-TB treatments that are effective for both drug-susceptible and drug-resistant TB, which continues to pose a serious challenge to the public health worldwide[12]. Interestingly, in our efforts to search for novel potential UBPs from Mtb, we identified a eukaryotic-like Ub-associated (UBA) domain-containing Mtb surface protein Rv1468c (PE_PGRS29), which belongs to the mycobacteria-specific PE_PGRS protein family. Rather than being ubiquitinated by E3 Ub ligases, Mtb Rv1468c was directly targeted by host Ub chains through UBA-dependent interaction, which led to the engulfment of mycobacteria into LC3-associated autophagosomes for Atg5-dependent autophagic clearance. Disruption of Rv1468c UBA domain to abolish its interaction with Ub impaired host xenophagic clearance of Mtb in macrophages, and elevated bacterial loads in mice with enhanced inflammatory responses. Our findings reveal a previously unrecognized role of Ub as an innate immune trigger that binds to the pathogen surface protein to initiate host antimicrobial autophagy, which process is independent of the conventional xenophagy pathway initiated by E3 Ub ligases-mediated ubiquitination of substrates from pathogenic bacteria or bacteria-containing vacuoles[1,9]. Our results also indicate a potentially important strategy adopted by Mtb to benefit its persistent intracellular infection through maintaining optimized intracellular bacterial loads to avoid excessive host inflammatory responses.

## Results

**Ub directly binds to mycobacterial surface**. Mtb was shown to cause disruption of phagosomal membranes for cytosolic access at a very early time of infection[13,14]. By using electron microscope, we did observe that some of Mtb were free in the cytosol of macrophages as early as 4 h post-infection (Supplementary Fig. 1a), characterized by being surrounded by none of the host membranes[15]. Host Ub was indicated to be associated with either the membranes of Mtb-containing phagosomes or the surface of mycobacteria accessing the cytosol for initiating antibacterial autophagy[3,5,16]. To determine the direct binding of Ub on Mtb surface, we used digitonin to selectively permeabilize the plasma membranes[3,17], and found that ~20% Mtb were colocalized with Ub in bone marrow-derived murine macrophages (BMDMs) (Supplementary Fig. 1b, c). When using digitonin plus Triton X-100 to permeabilize both plasma membranes and phagosomal membranes, the majority of Mtb (~80%) were detected by anti-Mtb antibody, and ~35% were colocalized with Ub (Supplementary Fig. 1b, c). These data indicate that during Mtb infection in macrophages, a considerable proportion of Ub directly adhered to bacterial surface. To further confirm that Ub could directly bind to mycobacterial surface, we next incubated Mtb cells with K63 or K48-linked poly-Ub chains (Ub[2–7]) in vitro, and then washed and lysed bacterial cells for immunoblot analysis. Intriguingly, we found that both K63 and K48 Ub chains could directly bind to Mtb in vitro, while little Ub chains bound to Mtb pretreated with proteinase K for degradation of bacterial surface-exposed proteins[18] (Fig. 1a, b). These results indicate that the mycobacterial surface protein plays a central role in mediating direct binding of Ub to Mtb. Consistently, confocal microscopy analysis further showed that a large proportion of Mtb was colocalized with either K63 or K48 Ub chains after incubation in vitro, which phenomenon was largely abolished by pretreating Mtb with proteinase K (Fig. 1c). Collectively, these experiments indicate that Ub could target to Mtb via direct interaction with unknown mycobacterial surface proteins in an E3 ligases-independent manner.

**Mtb Rv1468c binds to polyubiquitin via a UBA domain**. In our efforts to search for potential novel Ub-binding surface proteins in Mtb, we identified a UBA domain (32–66 amino acids, a typical UBD in eukaryotic cells) in Mtb Rv1468c using SMART (http://smart.embl.de)[19]. The eukaryotic UBAs contain a hydrophobic surface patch composed of a cluster of hydrophobic amino acids, which is indispensable for noncovalent interaction with Ub[20,21] (Supplementary Fig. 2a). To determine the direct binding of Rv1468c UBA with the specific Ub chains, we purified the GST-tagged wild-type (WT) Mtb Rv1468c and the truncated ΔUBA (67–370) and UBA (1–72) (Fig. 1d). In vitro precipitation analysis indicated that Rv1468c directly bound to K63 and K48 poly-Ub, but not mono-Ub, via the UBA domain (Fig. 1e). This tendency of Rv1468c UBA domain favoring poly-Ub chains is similar to many eukaryotic UBA domains[20,21]. We further found that Rv1468c could bind to all seven types of homotypic Ub chains (Supplementary Fig. 2b). Notably, there is no lysine in the sequence of Mtb Rv1468c that could be targeted for conventional ubiquitination by the E3 ligases[10]. We thus disrupted potentially key hydrophobic residues in Rv1468c UBA to examine the noncovalent interaction between Rv1468c and Ub (Supplementary Fig. 2a, c). Either mutation of V64 or L65 to hydrophilic glycine markedly reduced the strength of interaction between Rv1468c and Ub (Fig. 1f and Supplementary Fig. 2c). In HeLa cells, GFP-tagged V64G or L65G mutant had little colocalization with HA-tagged Ub compared to WT Rv1468c (Fig. 1g). Thus, the "VL" motif in Rv1468c UBA is critical for the binding to Ub.

BLAST search in UniProtKB_Bacteria database followed by phylogenetic analysis indicated that Rv1468c homologs were only present in mycobacterial species and highly conserved among the

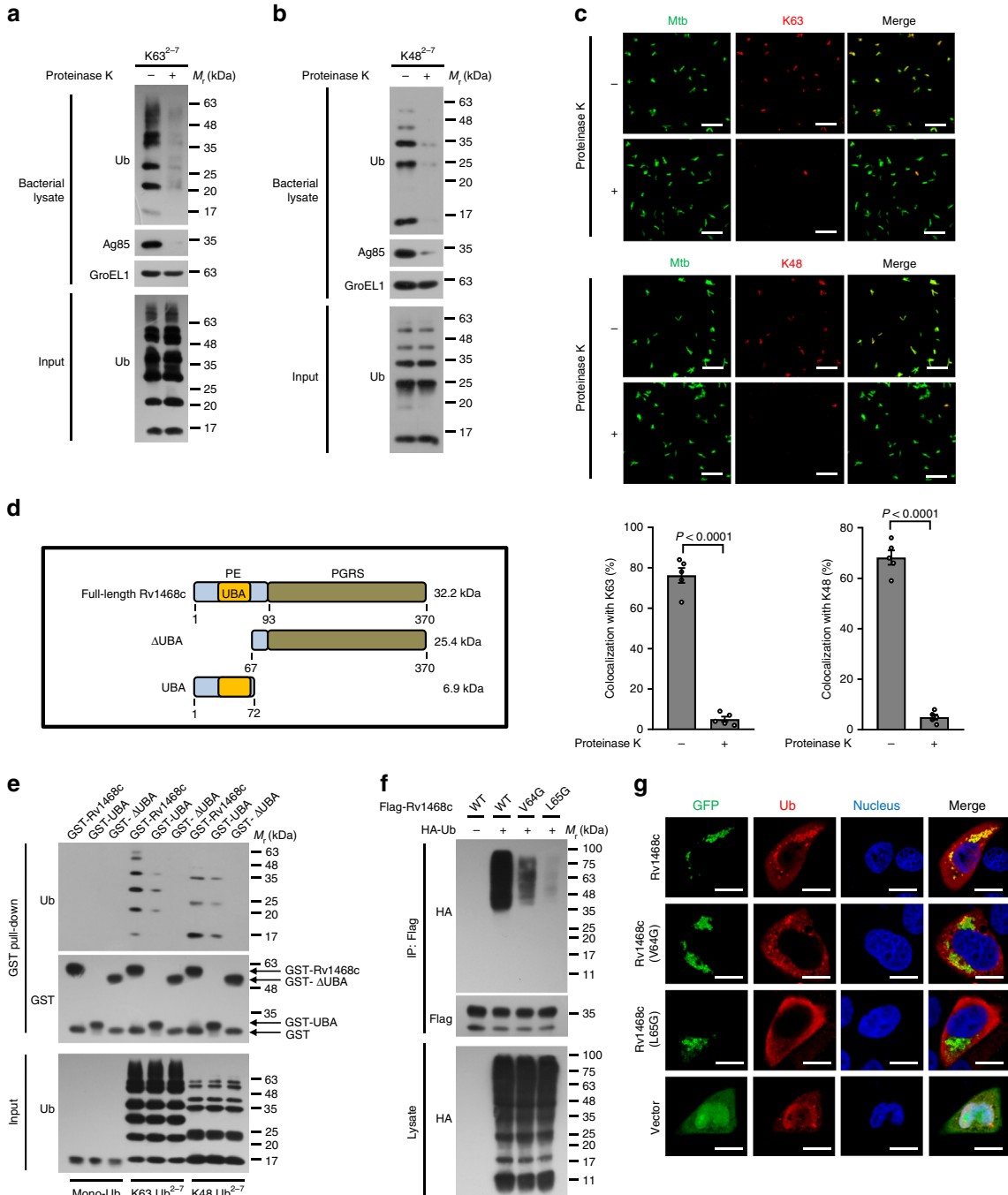

**Fig. 1** Mtb Rv1468c directly binds to poly-Ub chains in a UBA-dependent manner. **a**, **b** Immunoblot analysis of whole bacterial lysate of Mtb cells pretreated with (+) or without (−) proteinase K followed by incubation with K63$^{2-7}$ (**a**) or K48$^{2-7}$ (**b**) poly-Ub chains at 4 °C for 4 h. **c** Confocal microscopy analysis for colocalization of pSC301-Mtb stably expressing green fluorescent protein (GFP) with K63$^{2-7}$ or K48$^{2-7}$ Ub chains in vitro. Bacteria (green) treated with (+) or without (−) proteinase K were further incubated with poly-Ub chains as in **a** and **b**, and were then immunostained using anti-Ub antibody (red). Scale bars, 10 μm. The percent colocalizations of Ub and Mtb are shown at the bottom. A total of 100 bacterial cells were counted. Two-tailed unpaired *t*-test was performed. **d** Schematic representation of the Mtb Rv1468c constructs used. **e** In vitro interaction analysis of Ub and GST-tagged full-length Mtb Rv1468c or the truncated mutants as indicated in **d**. Purified recombinant GST-tagged Mtb Rv1468c variants were immobilized on glutathione sepharose resin. Mono-Ub (His$_6$-tagged), K63 or K48-linked poly-Ub (Ub$^{2-7}$) was added, and then the protein complexes were assayed by immunoblotting with indicated antibodies. **f** Immunoblot analysis of proteins immunoprecipitated (IP) with anti-Flag M2 Affinity Gel from lysates of HEK293T cells transfected with empty vector (−) or hemagglutinin (HA)-tagged Ub (+) and Flag-tagged wild-type (WT) Rv1468c or its mutants (V64G or L65G). **g** Confocal microscopy analysis of HeLa cells cotransfected with vectors encoding HA-tagged Ub (red) and GFP-tagged WT Rv1468c, V64G or L65G mutant (green). Nuclei (blue) were stained with the DNA-binding dye DAPI. Scale bars, 10 μm. Results are representatives from at least three independent experiments (mean ± s.e.m. of n = 5 in **c**). The source data used in **c** are provided in Source Data

pathogenic mycobacteria (Supplementary Fig. 2d). We further explored the diversity of Mtb gene *Rv1468c* in clinical isolates by using GMTV database (http://mtb.dobzhanskycenter.org)[22], and found only 1 strain with nonsynonymous mutation (A53T) in the UBA region among a total of 687 Mtb strains (Supplementary Fig. 2e and Supplementary Data 1), suggesting that Rv1468c UBA was relatively conserved. Taken together, these results indicate that Mtb Rv1468c could directly bind to poly-Ub chains via a unique UBA domain that harbored a conserved VL motif.

**Ub targets to Rv1468c on mycobacterial cell surface**. Mtb Rv1468c belongs to mycobacterial PE_PGRS family, which family proteins are predominantly surface-exposed in bacterial cell wall[23]. Therefore, we suspected that Mtb Rv1468c was also a bacterial surface protein which might be recognized by host Ub during infection. To test this idea, we deleted *Rv1468c* in Mtb H37Rv (Mtb Δ*Rv1468c*), and complemented Mtb Δ*Rv1468c* strain with WT *Rv1468c* (Mtb Δ*Rv1468c:Rv1468c*) or *Rv1468c* L65G (Mtb Δ*Rv1468c:Rv1468c* L65G) (Supplementary Fig. 3a, b). We first carried out cell fractionation assay of Mtb to characterize the subcellular localization of Rv1468c, and found that both WT Rv1468c and the L65G mutant were detected mostly in the cell wall fraction (Fig. 2a). As controls, Ag85 and GroEL1 were largely present in the cell wall fraction and cytoplasmic fraction, respectively[24] (Fig. 2a). We also deleted *Rv1468c* in *M. bovis* BCG (BCG Δ*Rv1468c*), or overexpressed Mtb Rv1468c in *M. smegmatis* (*Rv1468c-M. smegmatis*), a nontuberculous species that does not encode Rv1468c (Supplementary Data 2) for electron microscopy analysis to further confirm that Rv1468c was indeed expressed on the mycobacterial cell surface (Fig. 2b). We then investigated if Rv1468c could mediate the direct targeting of Ub to mycobacterial surface in vitro. After incubation of each mycobacterial strain with K63 or K48-linked Ub chains (Ub$^{2-7}$), we found that deletion of *Rv1468c*, or substitution of *Rv1468c* by L65G mutant in Mtb reduced the direct binding of Ub to mycobacteria (Fig. 2c, d). When Mtb cells were subjected to a short exposure to proteinase K before incubation, Rv1468c as well as the control protein Ag85, but not GroEL1, were markedly degraded, and the binding of Ub to Mtb was decreased (Fig. 2c, d). Confocal microscopy analysis also showed that deletion or L65G mutation of *Rv1468c* in Mtb markedly reduced the colocalizations of bacteria with K63 and K48 Ub chains in vitro (Supplementary Fig. 3c–f). These results indicate that Rv1468c UBA domain is exposed on the Mtb surface to mediate direct binding of Ub to Mtb. Consistently, when we incubated different *M. smegmatis* strains with K63 or K48-linked Ub chains (Ub$^{2-7}$) in vitro, we found that *Rv1468c-M. smegmatis* was able to bind more K63 and K48 Ub chains as compared to WT *M. smegmatis* or *M. smegmatis* expressing *Rv1468c* L65G (*Rv1468c* L65G-*M. smegmatis*) (Supplementary Fig. 4). These results further confirmed that Ub could indeed bind to Rv1468c UBA domain, rather than nonspecifically to the mycobacterial outer membrane.

*Rv1468c* mRNA was upregulated in WT Mtb after infection to macrophages (Supplementary Fig. 5a), indicating that Rv1468c was likely to be involved in Mtb-host interactions. Thus, we next investigated if Rv1468c participates in host Ub targeting to mycobacterial surface in vivo. To this end, BMDMs were infected with WT Mtb or Mtb Δ*Rv1468c* and then analyzed by electron microscope with immunogold labeling to trace host Ub and Mtb Rv1468c. Mtb cells displayed a triple layered cell envelope[25] (Fig. 3a, b), and Mtb Rv1468c was associated with host Ub on the surface of mycobacterial envelope in macrophages (Fig. 3a). Besides, we also found a marked reduction of Ub directly bound to the surface of Mtb Δ*Rv1468c* as compared to that of WT Mtb (Fig. 3b, c). Consistently, confocal microscopy analysis showed

that Mtb Δ*Rv1468c* and Mtb Δ*Rv1468c:Rv1468c* L65G had a decreased colocalization with Ub as compared to WT Mtb and Mtb Δ*Rv1468c:Rv1468c* in macrophages, which became more obvious overtime during infection (Fig. 3d, e). Furthermore, by using Ub linkage-specific antibodies, we found that deletion of *Rv1468c* or substitution of *Rv1468c* by L65G mutation in Mtb reduced ~2-fold of the colocalization of bacteria with either K63- or K48-linked Ub chains at 24 h post-infection (Supplementary Fig. 5b–e). Together, these results demonstrate that Rv1468c could mediate the direct targeting of host Ub to mycobacterial cell surface during the infection.

**Autophagic clearance of mycobacteria requires Rv1468c**. The Ub system could either participate in host immune defense against mycobacteria, or be exploited by mycobacteria to facilitate their intracellular survival[11]. Thus, we next determined the exact role of Rv1468c-Ub interaction at the interface of Mtb-host interactions. Mtb Δ*Rv1468c* and Mtb Δ*Rv1468c:Rv1468c* L65G had an enhanced viability than WT Mtb and Mtb Δ*Rv1468c: Rv1468c* in BMDMs (Fig. 4a), suggesting that Ub binding to Rv1468c was in favor of host immunity. Since Ub attachment to bacterial surface is a prime step for enhancing "eat-me" signals to recruit the host autophagy machinery[1,26], and Rv1468c was crucial for Ub binding to Mtb, we next investigated whether Rv1468c was involved in antimycobacterial autophagy. In macrophages that were activated by rapamycin, a widely used inhibitor of the mammalian target of rapamycin (mTOR) to induce autophagy[27,28], Mtb Δ*Rv1468c* exhibited more distinct advantage of intracellular survival than WT strain as compared to that in control group (Fig. 4b). On the contrary, when macrophages were treated by rapamycin along with 3-methyladenine, a phosphatidylinositol 3-kinase (PI3K) inhibitor that blocked the initiation of autophagy[27], the enhanced intracellular viability of Mtb Δ*Rv1468c* was largely abolished (Fig. 4b). Consistently, Mtb Δ*Rv1468c* had a significantly lower proportion of colocalization with endogenous LC3 and the lysosomal marker LAMP1 as compared to that of WT strain in BMDMs, and these differences became more obvious when macrophages were subjected to autophagic induction by rapamycin, interferon-γ (IFNγ), or amino acid and serum starvation[27,28] (Supplementary Fig. 6). However, when macrophages were treated with 3-methyladenine, the targeting of LC3 and LAMP1 to WT Mtb or Mtb Δ*Rv1468c* was similarly suppressed (Supplementary Fig. 6). Further experiments demonstrated that substitution of *Rv1468c* by the L65G mutant in Mtb had same effects as deletion of *Rv1468c* in Mtb on reducing the colocalization of bacteria with LC3 and LAMP1 in BMDMs (Fig. 4c–f). Thus, these results indicate that Ub targeting to the UBA domain of Rv1468c is crucial for host autophagic clearance of mycobacteria.

Since subversion of host xenophagy could also result from impaired ability of mycobacteria to cause phagosomal damage[2], which might be affected by deletion or mutation of *Rv1468c* in Mtb. Thus, to rule out the possibility that different degrees of Ub targeting for the WT and mutant Mtb strains could be due to differences in their ability to permeabilize the phagosomal membrane to gain access to the cytosol in macrophages, we performed confocal microscopy analysis of macrophages infected different Mtb strains for 4 h followed by selective permeabilization of the plasma membranes, and found that all strains exhibited similar accessibility to anti-Mtb antibody, indicating that they had comparable accessibility to the cytosol (Supplementary Fig. 7a, b). Consistently, by using electron microscope, we observed that all Mtb strains showed similar proportions of bacteria in the cytosol and in the phagosomes at 4 h post-infection in macrophages (Supplementary Fig. 7c, d). In addition,

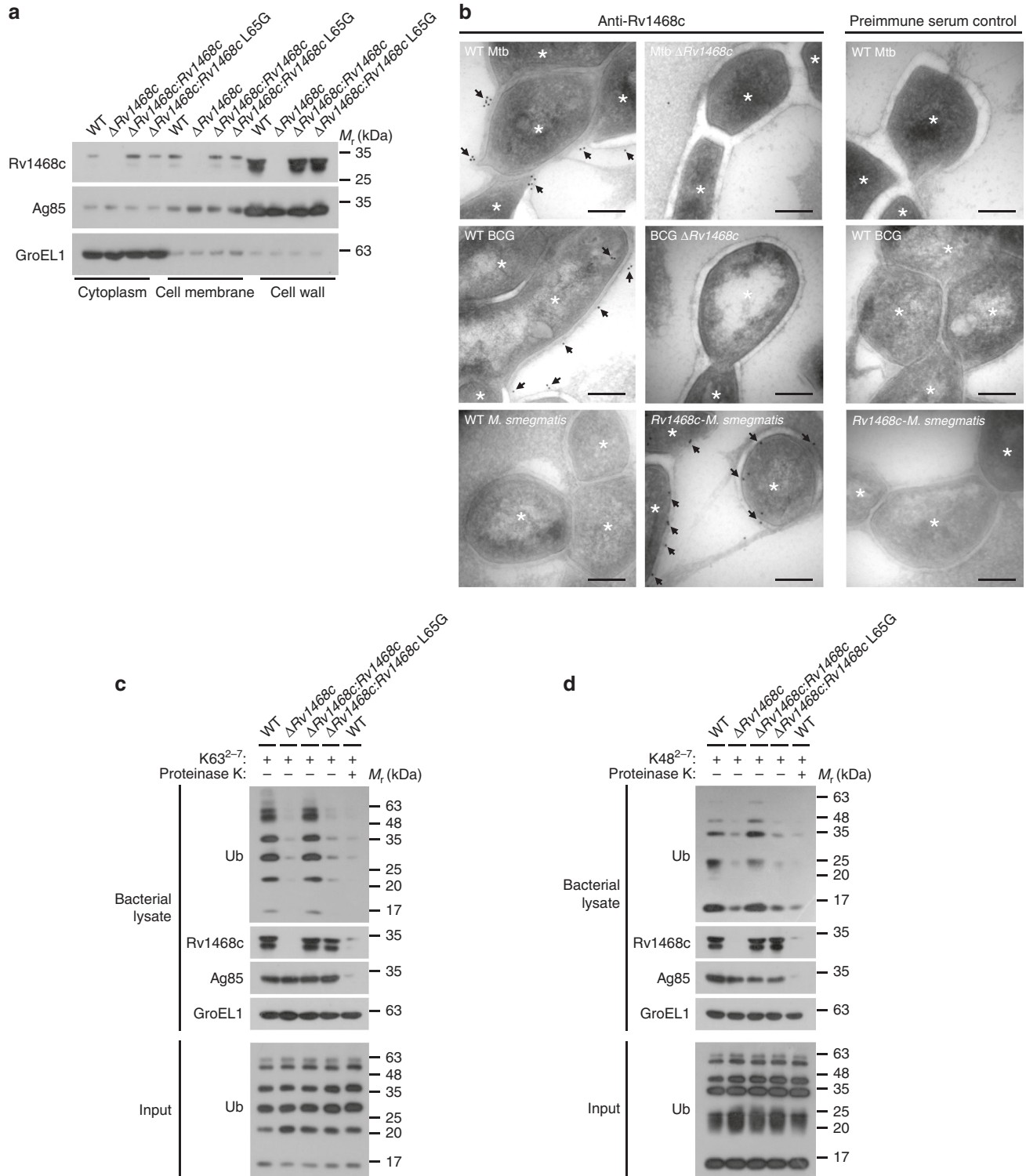

**Fig. 2** Rv1468c expresses on mycobacterial surface and mediates Ub binding to Mtb. **a** Immunoblot analysis of Rv1468c in subcellular fractions of the indicated Mtb strains. **b** Electron microscopy analysis for localization of Rv1468c immunolabelled with 10 nm gold (black arrows) on ultrathin sections of the mycobacterial strains as indicated. Staining controls for anti-Rv1468c were done with preimmune serum. Asterisks represent the mycobacterial cells. Scale bars, 100 nm. **c**, **d** Immunoblot analysis of whole bacterial lysate of the indicated Mtb strains pretreated with (+) or without (−) proteinase K followed by incubation with K63$^{2-7}$ (**c**) or K48$^{2-7}$ (**d**) poly-Ub chains (+) at 4 °C for 4 h. Results are representatives from at least three independent experiments

we also performed confocal microscopy analysis of infected macrophages to detect the colocalization of Mtb with galectin-3, an established indicator of damaged phagosomes caused by intracellular bacteria[29], and found that all Mtb strains had similar percentages of colocalization with galectin-3 (Supplementary Fig. 8), further confirming that Mtb-caused phagosomal disruption in macrophages was not affected by deletion or mutation of *Rv1468c*.

Next, we performed electron microscopy assay to further confirm the machinery of Rv1468c-dependent antimycobacterial

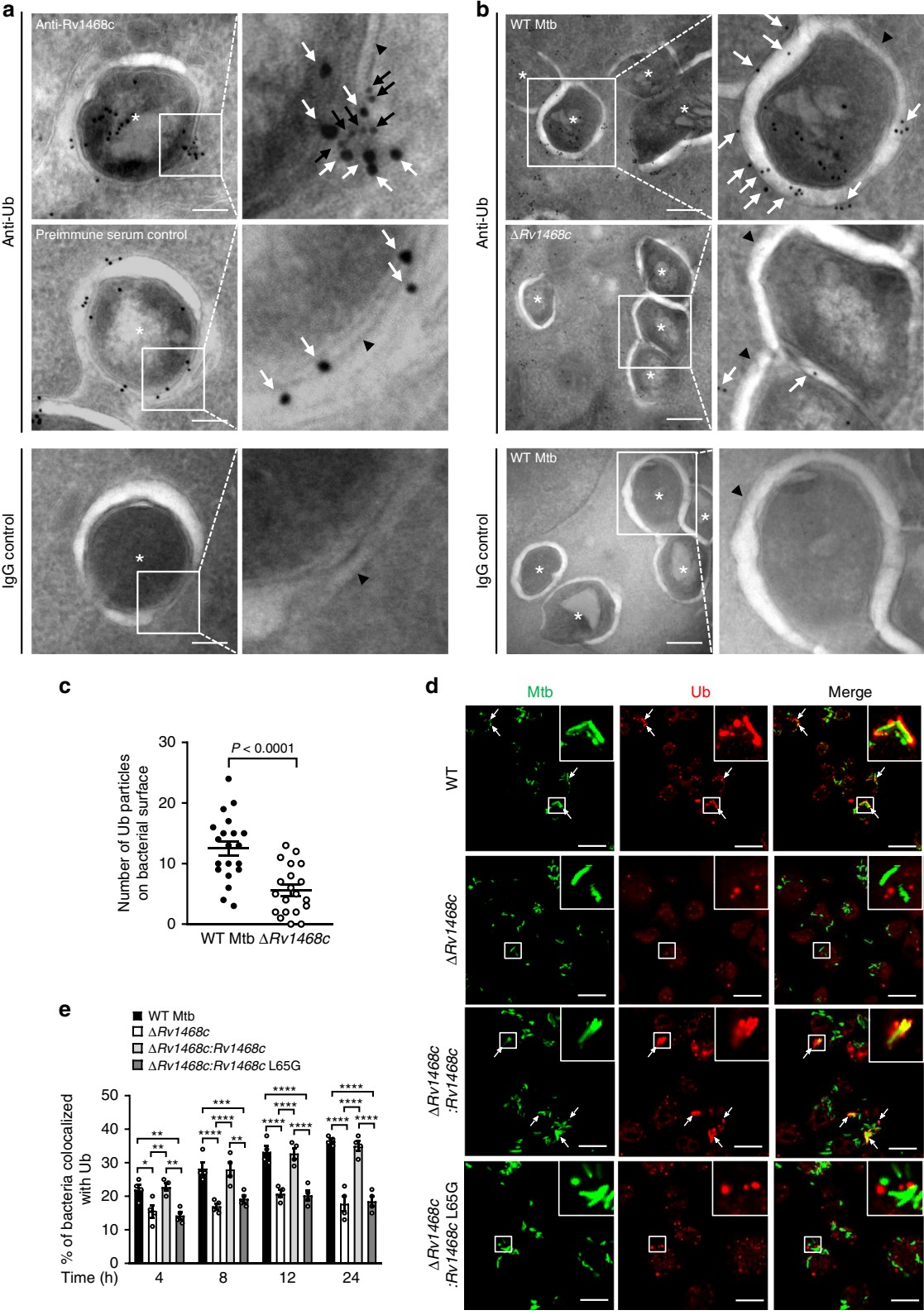

autophagy. As observed in LC3 immunogold staining, a considerable number of LC3 particles directly bound to the surface of WT Mtb, but not Mtb Δ*Rv1468c* (Fig. 5a, b), indicating that Rv1468c was required for host LC3 coating on the bacterial surface. Moreover, we also confirmed that Mtb could be targeted through macroautophagy pathway for degradation, as substantiated by the observation that ~30% bacilli were sequestered in

double or multiple-membraned vesicles in macrophages at 24 h post-infection[15,26,27,30] (Fig. 5c, d). The onion-like multilamellar structures that typically occurred during macroautophagy against intercellular pathogens were also frequently observed in WT Mtb-infected macrophages[27,30] (Fig. 5c). By comparison, deletion of *Rv1468c* in Mtb markedly reduced the proportion of bacteria present in autophagosomal vesicles but increased that in host

**Fig. 3** Rv1468c is required for Ub binding to Mtb during the infection in macrophages. **a** Electron microscopy analysis for bone marrow-derived murine macrophages (BMDMs) infected with WT Mtb strain. Cells were infected for 24 h, and were then immunolabelled for Rv1468c with 6 nm gold (black arrows) and for Ub with 10 nm gold (white arrows). Inserts (right panel) show representative areas on mycobacterial surface where Ub binds to Rv1468c. Scale bars, 100 nm. **b** Electron microscopy analysis for BMDMs infected with WT Mtb or Mtb ΔRv1468c as in **a**. Infected cells were immunolabelled for Ub with 10 nm gold (white arrows). Inserts (right panel) show representative areas where Ub binds to mycobacterial surface. Scale bars, 200 nm. Staining controls for anti-Rv1468c and anti-Ub were done with preimmune serum or mouse IgG1, respectively. Asterisks represent the mycobacterial cells. Arrowheads mark the bacterial wall. **c** Quantification of Ub immunogolds on mycobacterial surface in BMDMs treated as in **b**. A total of 20 bacterial cells were monitored for counting the numbers of Ub particles. Two-tailed unpaired $t$ test was performed. **d** Confocal microscopy analysis for colocalization of Mtb with Ub in BMDMs. BMDMs were infected with indicated Mtb strains at MOI = 5 for 24 h and were then immunostained using anti-Ub antibody (red). Bacteria (green) were prestained with Alexa Fluor 488 succinimidyl ester before infection. Arrows indicate the colocalization of bacteria with Ub. Inserts (enlarged views) show representative mycobacteria colocalized with Ub. Scale bars, 20 μm. **e** Percent colocalizations of Mtb with Ub in BMDMs treated as in **d** for 0–24 h. A total of 100 bacterial cells were counted. *$P < 0.05$; **$P < 0.01$; ***$P < 0.001$; ****$P < 0.0001$ (two-way ANOVA). Results are representatives from at least three independent experiments (mean ± s.e.m. of $n = 20$ in **c**; $n = 4$ in **e**). The source data used in **c** and **e** are provided in Source Data

cytosol (Fig. 5c, d), implying that Rv1468c was probably of importance for autophagic clearance of cytosolic bacilli whose surface was exposed for recognition by host Ub. Together, these data further confirmed that Rv1468c is indispensable for host clearance of mycobacteria via autophagy pathway.

**p62 is involved in Ub-Rv1468c interaction-mediated xenophagy.** Autophagy receptors including p62, NBR1, NDP52, and optineurin (OPTN) have been demonstrated to interact with both Ub on the protein substrates and LC3 on the phagophore during the xenophagy of intracellular pathogens[1,8,26]. Thus, we next determined whether these autophagy receptors mediated the interconnection of Ub-bound Rv1468c with LC3 to trigger the selective autophagy of mycobacteria. In HeLa cells, DsRed2-tagged Rv1468c were colocalized with both HA-tagged Ub and GFP-tagged p62, NBR1, NDP52 or OPTN, but Rv1468c (L65G) did not (Supplementary Fig. 9a). When Flag-tagged Rv1468c was coexpressing with GFP-tagged p62, NBR1, NDP52 or OPTN in HEK293T cells, it was able to coimmunoprecipitate with each of autophagy receptors in a UBA-dependent manner (Supplementary Fig. 9b, c). Since p62 exhibits the strongest affinity to Rv1468c and plays an indispensable role in multiple selective autophagy process by cooperation with other adaptors[31–33], we further performed confocal microscopy analysis to monitor p62 in macrophages during Mtb infection. As expected, deletion of Rv1468c or substitution of Rv1468c by L65G mutation in Mtb reduced the aggregation of p62 to the bacteria in BMDMs as compared to that of WT and Mtb ΔRv1468c:Rv1468c strains (Fig. 6). Therefore, p62 participates in mediating the Rv1468c-Ub interaction-triggered xenophagy.

**Deletion of p62 or Atg5 impairs Rv1468c-dependent xenophagy.** Atg5 was required for cytosolic LC3 (LC3-I) converting to lipidated LC3 (LC3-II) on the phagophores that initiated the formation of autophagosomes, and subsequently LC3-II and p62 underwent degradation along with cargoes when autophagy flux was unblocked[1,8,26]. Since Rv1468c-Ub interaction-triggered xenophagy was associated with the recruitment of p62 and LC3 to Mtb, we thus deleted p62 and Atg5 in RAW264.7 macrophages by using CRISPR/Cas9-mediated gene targeting to confirm that this antimycobacterial process was contributed by host selective autophagy, and we found that after infection for 24 h, Mtb ΔRv1468c and Mtb ΔRv1468c:Rv1468c L65G exhibited enhanced intracellular viability as compared to WT Mtb or Mtb ΔRv1468c: Rv1468c in WT RAW264.7 cells, but showed little survival advantage in p62$^{-/-}$ and Atg5$^{-/-}$ RAW264.7 cells (Supplementary Fig. 10a–c). Furthermore, deletion of p62 or Atg5 in RAW264.7 macrophages reduced the LC3 aggregation to mycobacteria as

well as the colocalization of LAMP1 with all strains (Supplementary Fig. 10d–g). Encouraged by the results from Atg5$^{-/-}$ RAW264.7 macrophages indicating that deletion of Atg5 could impair Rv1468c-dependent autophagic clearance of Mtb, we further generated Atg5$^{flox/flox}$-Lyz-Cre (Atg5$^{LyzM}$) mice that lacked Atg5 in myeloid cells[2,34,35] (Supplementary Fig. 11a, b), and obtained Atg5-deficient BMDMs derived from Atg5$^{LyzM}$ mice to further confirm the critical role of Atg5 in Rv1468c-Ub binding-dependent autophagy. Consistently, in the control BMDMs derived from Atg5$^{flox/flox}$ (Atg5$^{fl/fl}$) mice, but not Atg5$^{LyzM}$ BMDMs, Mtb ΔRv1468c and Mtb ΔRv1468c:Rv1468c L65G exhibited increased intracellular survival percentage, and exhibited more distinct advantage of viability when macrophages were stimulated by rapamycin (Fig. 7a). Deletion of Rv1468c or substitution of Rv1468c by using L65G mutant in Mtb also reduced the colocalization of bacteria with LC3 and LAMP1 in Atg5$^{fl/fl}$ BMDMs, but not in Atg5$^{LyzM}$ BMDMs (Fig. 7b–e). Furthermore, the colocalizations of WT Mtb and Mtb ΔRv1468c: Rv1468c, but not Mtb ΔRv1468c or Mtb ΔRv1468c:Rv1468c L65G, with Ub increased in Atg5$^{LyzM}$ BMDMs, as compared to Atg5$^{fl/fl}$ BMDMs, which might be explained by the possibility that host autophagy flux for degradation of Ub-bound bacteria was blocked by Atg5 deficiency (Supplementary Fig. 11c, d). Taken together, these data indicate that disturbance of host autophagy pathway by deletion of p62 or Atg5 impairs Rv1468c-Ub interaction-triggered clearance of mycobacteria.

**Rv1468c-dependent clearance of mycobacteria requires Atg5.** To further explore the role of Atg5 in Rv1468c-Ub interaction-triggered host immune defense against mycobacteria, we challenged Atg5$^{fl/fl}$ and Atg5$^{LyzM}$ mice with WT and various mutant Mtb H37Rv strains. Notably, deletion of Rv1468c or substitution of Rv1468c with L65G mutant in Mtb increased bacterial loads in the lungs of the control mice, but not the Atg5-deficient mice, at 2 and 3 weeks post-infection (Fig. 8a). Histopathological assay of lungs also showed that Atg5$^{fl/fl}$ mice, but not Atg5$^{LyzM}$ mice, infected with Mtb ΔRv1468c or Mtb ΔRv1468c:Rv1468c L65G had larger lesions with increased total cellular and neutrophilic infiltration, as well as more abundance of acid-fast bacilli in lung sections than those infected with WT Mtb and Mtb ΔRv1468c: Rv1468c at 3 weeks post-infection (Fig. 8b, c). Moreover, by 3 weeks post-infection, Atg5$^{fl/fl}$ mice infected with Mtb ΔRv1468c or Mtb ΔRv1468c:Rv1468c L65G exhibited increased cellular infiltration in livers as compared to WT Mtb and Mtb ΔRv1468c: Rv1468c; while Atg5$^{LyzM}$ mice infected with each of Mtb strains had a similar histopathological change in the livers (Fig. 8d). Consistently, quantitative PCR analysis further detected increased mRNA levels of pro-inflammatory cytokines including Tnf, Il1b

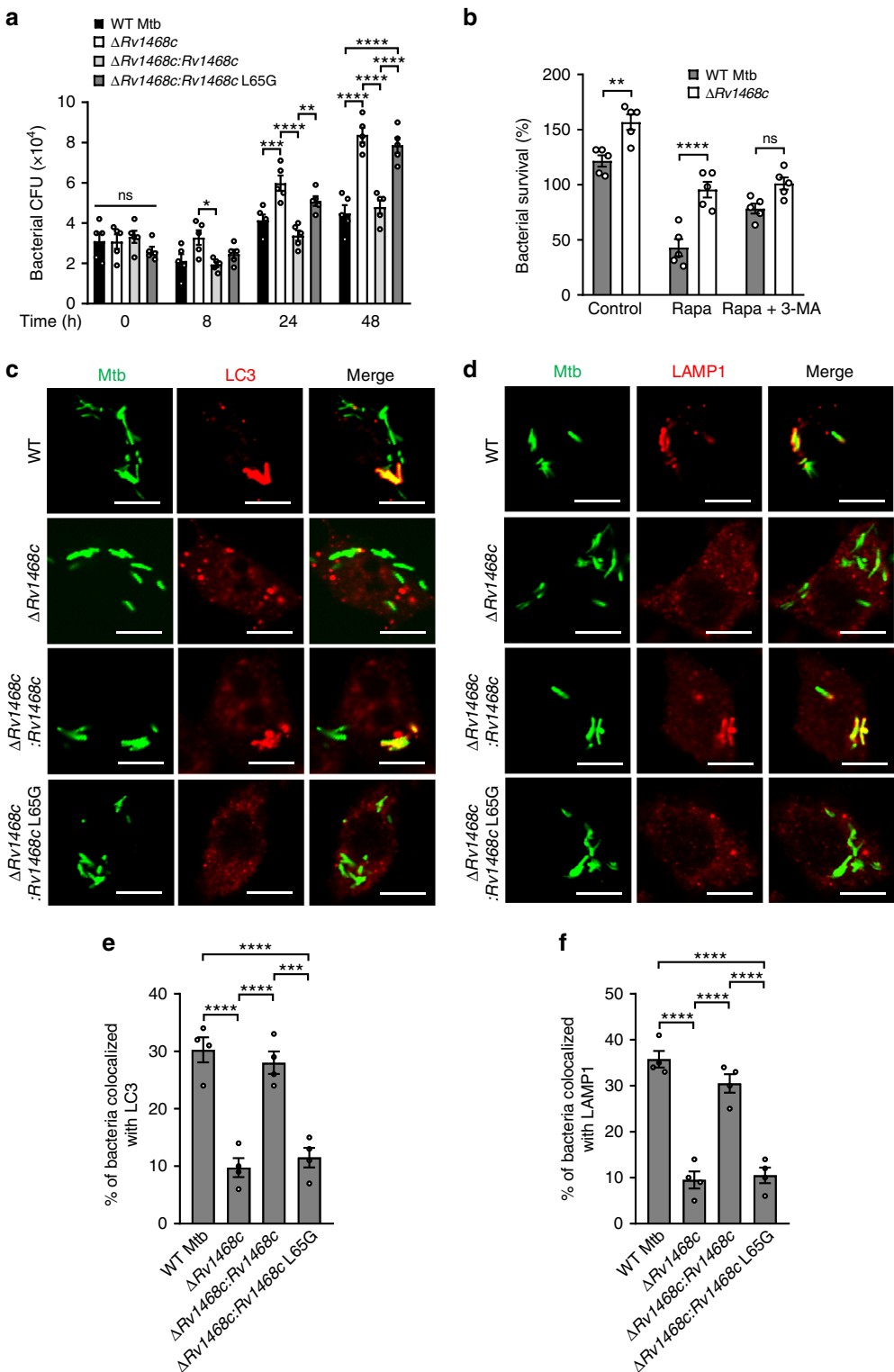

**Fig. 4** Deletion of *Rv1468c* promotes intracellular survival of Mtb and reduces colocalizations of Mtb with LC3 and LAMP1. **a**, **b** Survival of Mtb in BMDMs. Cells were infected with the indicated Mtb strains at MOI = 1 for 0–48 h (**a**), or 24 h after being treated with 50 μM rapamycin (Rapa), 50 μM Rapa along with 10 mM 3-methyladenine (3-MA), or equal volume of DMSO (Control) for 4 h (**b**). *P* > 0.05, not significant (ns); ***P* < 0.05; ****P* < 0.01; *****P* < 0.001; ******P* < 0.0001 (two-way ANOVA). **c**, **d** Confocal microscopy analysis for colocalizations of Mtb with LC3 or LAMP1. BMDMs were infected with each of the indicated bacterial strains at MOI = 5 for 24 h and were then immunostained using anti-LC3 (**c**) or anti-LAMP1 (**d**) antibody (red). Bacteria (green) were prestained with Alexa Fluor 488 succinimidyl ester before infection. Scale bars, 5 μm. **e**, **f** Percent colocalization of the indicated Mtb strains with LC3 (**e**) or LAMP1 (**f**) in macrophages infected as in **c** and **d**. A total of 100 bacterial cells were counted. *****P* < 0.0001 (one-way ANOVA). Results are representatives from at least three independent experiments (mean ± s.e.m. of *n* = 5 in **a**, **b**; *n* = 4 in **e**, **f**). The source data used in **a**, **b**, **e**, **f** are provided in Source Data

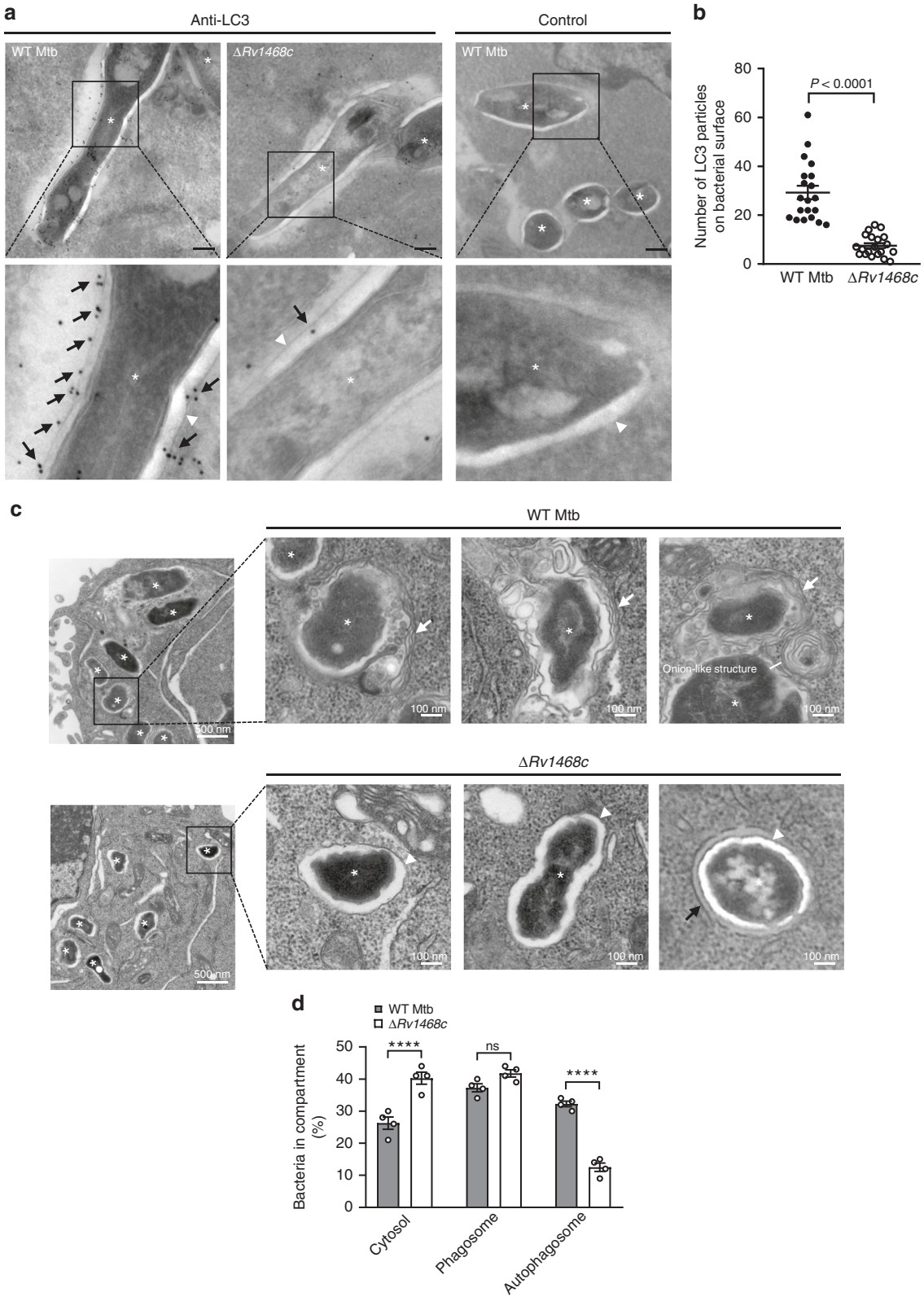

and *Il6* in the lungs of *Atg5<sup>fl/fl</sup>* mice infected with Mtb Δ*Rv1468c* or Mtb Δ*Rv1468c:Rv1468c* L65G, as compared to that infected with WT Mtb or Mtb Δ*Rv1468c:Rv1468c* at 3 weeks post-infection (Supplementary Fig. 12). The enhanced immuno-pathology in *Atg5<sup>fl/fl</sup>* mice caused by disruption of *Rv1468c* in Mtb was probably due to increased bacterial loads that triggered

stronger host inflammatory responses at 3 weeks post-infection. While the deletion of *Atg5* in myeloid cells impaired the host autophagic functionality, leading to increased susceptibility to mycobacterial infection with excessive inflammatory responses in *Atg5<sup>LyzM</sup>* mice infected with all Mtb strains, which results were consistent with data from previous studies[2,35]. Together, these

**Fig. 5** Deletion of *Rv1468c* in Mtb reduces autophagy targeting to mycobacteria. **a** Electron microscopy analysis for LC3 targeting on the surface of Mtb in BMDMs. BMDMs were infected with WT Mtb or Mtb Δ*Rv1468c* for 24 h and were then immunolabelled for LC3 with 10 nm gold (black arrows). Staining controls for anti-LC3 were done without primary antibody. Asterisks represent the mycobacterial cells. Arrowheads mark the bacterial wall. Scale bars, 200 nm. **b** Quantification of LC3 immunogolds on mycobacterial surface in BMDMs treated as in **a**. A total of 20 bacterial cells were monitored for counting the numbers of LC3 particles. Two-tailed unpaired *t*-test was performed. **c** Electron microscopy analysis of Mtb subcellular localizations within BMDMs. BMDMs were infected with WT Mtb or Mtb Δ*Rv1468c* as in **a** and were then processed for electron microscopy. Onion-like structure was observed in WT Mtb-infected BMDMs. White arrows indicate the membranes of autophagosomal vacuoles. The black arrow indicates the phagosomal membrane. Arrowheads indicate bacterial walls. Asterisks represent the mycobacterial cells. **d** Quantification of Mtb subcellular localizations within BMDMs treated as in **c**. A total of 100 bacterial cells were counted. *P* > 0.05, not significant (ns); ****P* < 0.0001 (one-way ANOVA). Results are representatives from at least three independent experiments (mean ± s.e.m. of *n* = 20 in **b** and *n* = 4 in **d**). The source data used in **b** and **d** are provided in Source Data

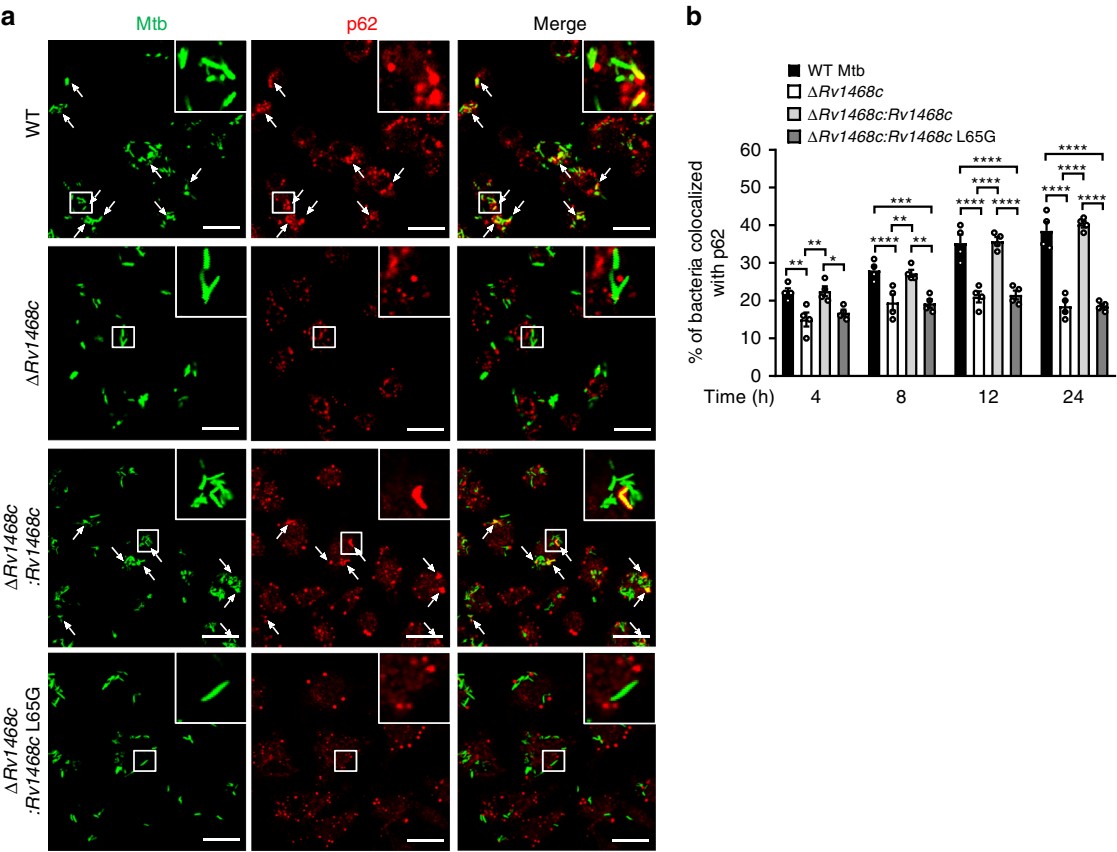

**Fig. 6** Rv1468c is required for host p62 targeting to Mtb in macrophages. **a** Confocal microscopy analysis for colocalization of Mtb with p62 in BMDMs. BMDMs were infected with the indicated Mtb strains at MOI = 5 for 24 h and were then immunostained using anti-p62 antibody (red). Bacteria (green) were prestained with Alexa Fluor 488 succinimidyl ester before infection. Arrows indicate the colocalization of bacteria with p62. Inserts (enlarged views) show representative mycobacteria colocalized with p62. Scale bars, 20 μm. **b** Percent colocalization of Mtb with Ub in BMDMs treated as in **a** for 0–24 h. A total of 100 bacterial cells were counted. **P* < 0.05; ***P* < 0.01; ****P* < 0.001; *****P* < 0.0001 (two-way ANOVA). Results are representatives from three independent experiments (mean ± s.e.m. of *n* = 4 in **b**). The source data used in **b** are provided in Source Data

data indicate that Atg5 is required for Ub-Rv1468c interaction-mediated host immune clearance of mycobacteria. Collectively, this study reveals a novel host xenophagy mechanism that initiated by direct binding of Ub to mycobacterial surface protein Rv1468c (Supplementary Fig. 13).

## Discussion
Autophagy has been well-documented for its role as a host immune defense against intracellular pathogens including Mtb[2–4,26,35]. Disruption of host xenophagy pathway by deletion of core autophagy genes such as *Atg5* and *Atg7* (refs. [2,35,36]), E3

ligases genes *Park2* and *Smurf1* (refs. [3,4]), autophagy receptor genes *p62* and *NDP52* (refs. [2,5]), or other selective autophagy-associated genes such as cyclic GMP-AMP synthase (cGAS)[37,38], *Sting*[2], *Tbk1*[2] and *IRGM*[39], could facilitate mycobacterial survival in macrophages and in mice. In this study, we found that various autophagy receptors including p62, NBR1, NDP52, and OPTN were all able to interact with Rv1468c, a UBA domain-containing Mtb surface protein. Interestingly, deletion of *p62* in macrophages effectively blocked Rv1468c-Ub interaction-mediated autophagy of Mtb, which finding is consistent with the previous notion that those autophagy receptors likely perform non-redundant functions and probably cooperate to mediate autophagy during

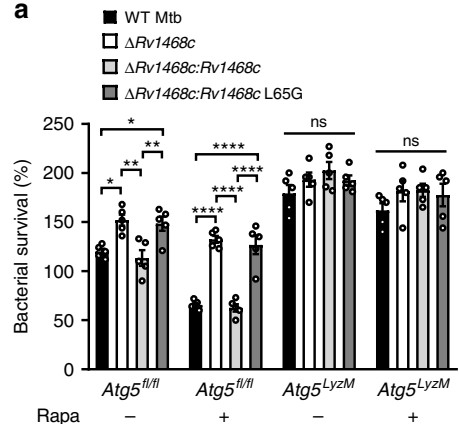

**Fig. 7** Deletion of *Atg5* suppresses Rv1468c-triggered autophagic clearance of Mtb. **a** Survival of Mtb in macrophages. BMDMs derived from *Atg5^flox/flox* (*Atg5^fl/fl*) or *Atg5^flox/flox*-Lyz-Cre (*Atg5^LyzM*) mice were pretreated with ( + ) or without (–) 50 μM rapamycin (Rapa) for 4 h to induce autophagy, and were then infected with each of the indicated Mtb strains at MOI = 1 for 24 h. $P > 0.05$, not significant (ns); *$P < 0.05$; **$P < 0.01$; ****$P < 0.0001$ (two-way ANOVA). **b**, **c** Confocal microscopy for colocalizations of Mtb with LC3 or LAMP1 in BMDMs. *Atg5^fl/fl* or *Atg5^LyzM* BMDMs were infected with the indicated Mtb strains for 24 h, and were then immunostained using anti-LC3 (**b**) or anti-LAMP1 (**c**) antibody (red). Bacteria (green) were prestained with Alexa Fluor 488 succinimidyl ester before infection. Scale bars, 10 μm. **d**, **e** Percent colocalizations of the indicated bacterial strains with LC3 (**d**) or LAMP1 (**e**) in *Atg5^fl/fl* or *Atg5^LyzM* BMDMs treated as in **b** and **c**. A total of 100 bacterial cells were counted. $P > 0.05$, not significant (ns); ****$P < 0.0001$ (two-way ANOVA). Results are representatives from at least three independent experiments (mean ± s.e.m. of $n = 5$ in **a**; $n = 4$ in **d**, **e**). The source data used in **a**, **d**, **e** are provided in Source Data

mycobacterial infection[8]. We also found that deletion of *Atg5* in macrophages almost completely subverted the Rv1468c-triggered autophagic clearance of mycobacteria, and mice lacking Atg5 in myeloid cells also failed to exert Rv1468c-elicited host immune defense against mycobacteria. These findings are consistent with previous observations supporting an essential role of Atg5 in protecting host against mycobacterial infection via its autophagy-related functions[2,26,35]. There was also a study showing that Atg5 could modulate neutrophil-mediated pathologic inflammation to control Mtb infection[40]. In the meantime, it was shown that host deficient in other autophagy-related genes (such as *Atg3*, *Atg5*, *Atg7*, *Atg14*, *Atg16l1*, and *Beclin 1*) generally results in hyperinflammatory syndromes during the infection[35,40–43]. Thus, there exist complicated mutual effects between autophagy and inflammation during host-pathogen interactions.

In eukaryotic cells, Ub signaling could be decoded by various Ub receptors that contain different UBDs for hydrophobic interaction with Ub, which contributes to the multifaceted regulatory roles of the Ub system in diverse biological processes including pathogen-host interactions[1,10]. In this study, we found that Mtb surface protein Rv1468c harbors a eukaryotic-type UBA domain, a typical UBD in eukaryotic cells that have not been previously identified in bacterial pathogens. The UBA domain of Rv1468c could be targeted by host Ub independent of E3 Ub ligases, thus mediating the aggregation of Ub to the bacteria for eliciting antibacterial autophagy. Based on our data from in vitro binding assays, Mtb Rv1468c is able to bind to free-floating poly-Ub chains, which Ub chains in eukaryotic cells are emerging as key factors at the interface of the host-pathogen interactions[44–46]. Nevertheless, since both free-floating and substrate-attached poly-Ub chains exist within host cells, we do not exclude the possibility that a portion of Rv1468c may also interact with certain substrate-attached poly-Ub chains during Mtb infection in macrophages. Previous studies have demonstrated that E3 ligases Parkin and Smurf1 specifically mediate K63 and K48 ubiquitination of Mtb-associated structures, respectively[3,4]. While in this study, we found that Rv1468c could interact with all seven types of homotypic Ub chains. Moreover, given the fact that no lysine site is present in Rv1468c for conventional ubiquitination by the E3 ligases, together with our finding that it could bind to the Ub chains in absence of the E3 ligases in vitro, our data indicate that Rv1468c is not likely to be as a substrate of Parkin or Smurf1, but rather an ubiquitination-independent Ub-binding target on mycobacterial surface. In macrophages, the E3 ligases-dependent ubiquitination of Mtb-containing phagosomes could occur as early as 4 h post-infection[3]. As the infection continues, increasing number of Mtb starts to damage the phagosomes and escape into the cytosol[13,14,16], where the host Ub might target the unsheltered mycobacteria by direct binding to Rv1468c on their surface. Therefore, we propose that both Rv1468c UBA-mediated direct binding of Ub to the mycobacterial surface and E3 ligases-mediated ubiquitination of Mtb-associated structures are involved in Ub-mediated antimycobacterial autophagy during the infection.

Initially, it seems counterintuitive that Mtb would retain such a conserved UBA domain-containing protein like Rv1468c that functions to mark bacteria for host clearance and thus avoiding excessive host inflammation. In fact, the long-term intimate interplays between Mtb and the host are quite nuanced, rather than irreconcilably in conflict until one is defeated[47]. In about 90% of the infected individuals, Mtb could persist in the host by establishing a latent state instead of causing host severe inflammatory responses and tissue damage, as many severe and acute infection-causing pathogens normally do[48]. Consistently, current evidence indicates that Mtb strains with relatively attenuated virulence may be better tolerated by patients for long-term, which may ultimately lead to an extended duration of illness and increased Mtb transmission potential[49,50]. There is also evidence suggesting that the slowly-replicating phenotypes of Mtb could help maintain the bacterial population in the host by continuously adapting to dynamic microenvironments in granulomas for prolonged infection[51]. Furthermore, another study suggests that lower induction of pro-inflammatory cytokines in infected individuals might be a contributing factor to the evolutionary success of modern Beijing strains of Mtb, as compared to the ancient strains[52]. Thus, Rv1468c UBA domain-triggered autophagy could be a viable evolutionary strategy adopted by Mtb to maintain long-term intracellular survival through self-controlling its intracellular bacterial loads to avoid excessive host inflammatory immune responses, which might favor prolonged infection and greater transmission. Interestingly, proteomic profiling of Mtb revealed that Rv1468c is expressed both at the early and chronic stages of infection[53], and the expression of Rv1468c could be regulated by IS*6110* in Mtb[54], further suggesting that the expression and retention of this protein might somehow exert certain beneficial advantages to the pathogen. Probably, Rv1468c-triggered autophagy might help preserve indolent growth of Mtb in vacuolar compartments, since the establishment of a sheltered niche within vacuoles mimicking normal cellular compartments could effectively elude host immune surveillance and pathogen clearance[55]. This notion could be supported by a variety of studies demonstrating that the ability to create and maintain a specialized vacuolar organelle that supports bacterial replication is an important survival strategy for many intracellular pathogens such as Mtb, *Legionella pneumophila* and *Brucella abortus*[56–58]. On the other side, the timely removal of cytosolic bacteria via Rv1468c-triggered autophagy could probably minimize the cytosolic exposure of microbial patterns and balance the intracellular bacterial burden to avoid host excessive inflammatory responses[59,60]. Nevertheless, the overall effect of the Rv1468c-triggered autophagy on the ultimate fate of intracellular Mtb may depend upon multiple factors, including whether the xenophagy flux, the complete process of xenophagy in which pathogens contained within autophagosomes are digested by lysosomes, is compromised or not in Mtb-infected macrophages[61]. Thus, the identification of a eukaryotic-type domain such as UBA in a Mtb surface protein for host immune recognition strengthened the notion that the intimate host-pathogen interactions drive their

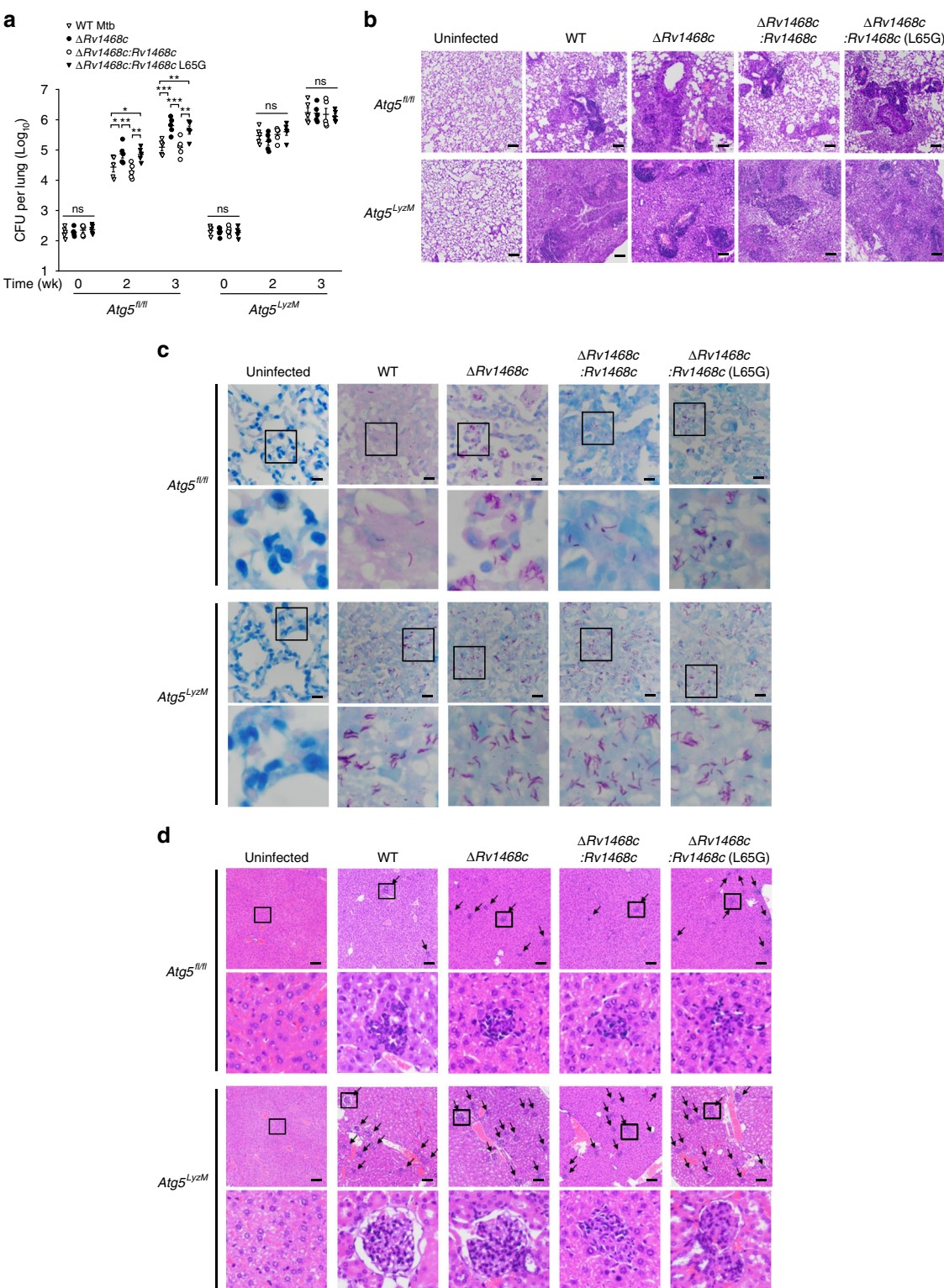

**Fig. 8** Atg5 is required for Rv1468c-triggered host immune clearance of Mtb in Mice. **a** Bacterial titers in homogenates of lungs from $Atg5^{flox/flox}$ ($Atg5^{fl/fl}$) or $Atg5^{flox/flox}$-Lyz-Cre ($Atg5^{LyzM}$) mice intratracheally infected with $1.0 \times 10^5$ CFU of the indicated Mtb strains for 0–3 weeks. $P > 0.05$, not significant (ns); *$P < 0.05$; **$P < 0.01$; ***$P < 0.001$ (mean ± s.e.m. of $n = 5$ mice per group; two-way ANOVA). **b** Histopathology of lung sections. Scale bars, 100 μm. **c** Acid-fast staining of lung sections. Scale bars, 10 μm. **d** Histopathology of liver sections. Arrows indicate foci of cellular infiltration. Scale bars, 100 μm. In **c** and **d**, the bottom panels show enlarged views of boxed areas. For **b**–**d**, $Atg5^{fl/fl}$ or $Atg5^{LyzM}$ mice infected with the indicated Mtb strains as in **a** for 3 weeks. Results are representatives from three independent experiments. The source data used in **a** are provided in Source Data

dynamic coevolution and antagonism, which regulate the diverse outcomes of pathogen persistence and host resistance[50].

Mtb encodes five ESAT6 secretion complexes (ESX-1 to ESX-5) known as type VII secretion systems[62]. ESX-1 is the most studied secretion system that shown to be involved in various host-pathogen interactions[2,62]. Notably, Rv1468c belongs to myco-bacterial PE_PGRS family, which family proteins has been demonstrated to be secreted by ESX-5, another secretion system of importance for Mtb pathogenicity without fully understanding of the molecular mechanisms[23,63–65]. Intriguingly, blocking PE_PGRS secretion in Mtb or *M. marinum*, a natural pathogen of ectotherms with close genetic relative of the *Mycobacterium tuberculosis* complex (MTBC), also leads to increased bacterial loads and inflammation in the host[64,65], suggesting that the ESX-5 system and its substrates of PE_PGRS proteins, may play important roles in triggering host immune defense and mod-ulating mycobacterial virulence for persistent intracellular survival[62,64,65]. PE_PGRS family proteins are mostly restricted to MTBC and are predominately expressed on the mycobacterial cell surface for direct interactions with the host[23]. It is indicated that PE_PGRS proteins might serve as a major source of myco-bacterial antigens[23,66], and the large majority of human T cell epitopes are confined to the relatively conserved PE domains, rather than the highly variable PGRS domains, of the PE_PGRS proteins[66], which is consistent with our finding that the Ub-targeting UBA domain, which is located in the PE domain of Rv1468c, is relatively conserved. The sequence conservation of the Ub-targeting and xenophagy-triggering UBA domain in Rv1468c, together with evidences indicating that autophagic induction could enhance the efficacy of TB vaccines by promoting mycobacterial antigen presentation[28,67], suggest that Rv1468c may serve as a potentially ideal molecular target for developing novel anti-TB therapies, especially when combined with xeno-phagy flux-modulating agents. Nonetheless, as noted earlier, there is a possibility that Rv1468c might be manipulated by the pathogen under certain circumstances to its own advantage, thus caution need be taken and both benefits and potential unfavor-able effects should be considered while choosing to target Rv1468c for TB treatment.

In summary, our findings reveal a novel host antimicrobial xenophagy pathway triggered by direct binding of Ub to a mycobacterial surface protein, indicating a previously unrecog-nized role of Ub in host innate immune defense against invading pathogens. Our findings also help expand our understanding of the intricate and dynamic interactions between the host and the invading pathogens.

## Methods

**Bacterial strains and plasmids.** *E. coli* DH5α and BL21 were grown in flasks using LB medium for genetic manipulations or protein overexpression. *M. smegmatis*, *M. bovis* BCG and *M. tuberculosis* (Mtb) H37Rv strains were grown in Middlebrook 7H9 broth (7H9) supplemented with 10% oleic acid-albumin-dextrose-catalase (OADC) and 0.05% Tween-80 (Sigma), or on Middlebrook 7H10 agar (BD) sup-plemented with 10% OADC. The BCG and Mtb H37Rv strain with deletion of the gene encoding Rv1468c (BCG Δ*Rv1468c* and Mtb Δ*Rv1468c*) was created by use of pJV53 system[68]. pMV306 plasmid (provided by W.R. Jacobs, Albert Einstein College of Medicine, Yeshiva University) was used to complement the strain Mtb Δ*Rv1468c* with WT *Rv1468c* or create variant strain Mtb Δ*Rv1468c* L65G. The recombinant *M. smegmatis* strains overexpressing Mtb Rv1468c or its mutant Rv1468c (L65G) were created by using mycobacterium shuttle vector pMV261 (provided by W.R. Jacobs, Albert Einstein College of Medicine, Yeshiva Uni-versity). The recombinant Mtb H37Rv strains stably expressing GFP (pSC301-Mtb) was created by using vector pSC301 (Addgene; Cat# 31851). The mammalian expression plasmid for HA-tagged ubiquitin (HA-Ub) was provided by F. Shao (National Institute of Biological Sciences, Beijing), and HA-K6 only, HA-K11 only, HA-K27 only, HA-K29 only, HA-K33 only, HA-K48 only and HA-K63 only ubiquitin were provided by H. Wang (Xiamen University, Xiamen). The full-length cDNA of human p62, NBR1, NDP52 or OPTN were amplified from HEK293T cDNA. For expression in mammalian cells, genes were cloned into the vector pEGFP-N1, pDsRed2-N1 or p3 × Flag-CMV14. Prokaryotic expression plasmids

were constructed by inserting the genes into the vector pGEX-6p-1 or pET30a. Point mutations were generated by using Fast Mutagenesis Kit V2 (Vazyme Bio-tech). All of the strains, plasmids, and primers used in this study were detailed in Supplementary Data 2.

**Antibodies and reagents.** Rabbit anti-Rv1468c antibody was produced and pur-ified by GenScript Biotechnology[11]. Briefly, a total of 5 mg of His$_6$-tagged Rv1468c protein was purified and solubilized in Freund's complete adjuvant for injection into rabbits. The antibody specific to Rv1468c was isolated by passaging the immunized rabbit serum on protein A agarose (#sc-2001; Santa Cruz). The fol-lowing antibodies were used in this study: anti-GFP (#ab1218, Abcam, 1:3000 for immunoblot analysis), anti-Flag (#F3165, Sigma-Aldrich, 1:4000 for immunoblot analysis), anti-HA (#3724, CST, 1000 for immunoblot analysis and 1:200 for immunofluorescence microscopy), mouse IgG1 (#ab18443, Abcam, 1:100 for immunofluorescence microscopy and 1:10 for immunoelectron microscopy), anti-ubiquitin (#13-1600, Invitrogen, 1:1000 for immunoblot analysis, 1:100 for immunofluorescence microscopy, and 1:10 for immunoelectron microscopy), anti-ubiquitin specific for K63 (#ab179434, Abcam, 1:400 for immunofluorescence microscopy), anti-ubiquitin specific for K48 (#05–1307, Millipore, 1:400 for immunofluorescence microscopy), anti-LC3 (#L7543, Sigma-Aldrich, 1:3000 for immunoblot analysis, 1:400 for immunofluorescence microscopy, and 1:50 for immunoelectron microscopy), anti-LAMP1 (#ab24170, Abcam, 1:200 for immu-nofluorescence microscopy), anti-Galectin-3 (#ab2785, Abcam, 1:200 for immu-nofluorescence microscopy), anti-p62 (#NBP1–48320, Novus Biologicals, 1:200 for immunofluorescence microscopy), anti-GAPDH (#sc-25778, Santa Cruz, 1:4000 for immunoblot analysis), anti-Tubulin (#T5168, Sigma-Aldrich, 1:4000 for immunoblot analysis), anti-GST (#TA-03, ZSGB-BIO, 1:3000 for immunoblot analysis), anti-GroEL1 (#NBP2–32867, Novus Biologicals, 1:1000 for immu-noblot analysis), anti-Ag85 (#ab36731, Abcam, 1:1000 for immunoblot analysis) and anti-Mtb (#ab905, Abcam, 1:400 for immunofluorescence microscopy). Murine IFNγ (#11276905001) was purchased from Sigma-Aldrich. Rapamycin (#S1039) and 3-methyladenine (#ab120841) were purchased from Selleck and Abcam, respectively. K63-linked Poly-Ub chains (Ub$^{2–7}$) (#UC-330) and K48-linked Poly-Ub chains (Ub$^{2–7}$) (#UC-230) were purchased from Boston Biochem.

**Cell culture and transfection.** HEK293T cells (ATCC CRL-3216), HeLa cells (ATCC CCL-2) and RAW264.7 cells (ATCC TIB-71) were obtained from the American type culture collection (ATCC). Cells were cultured in Dulbecco's modified Eagle's medium (DMEM; Gibco) with 10% fetal bovine serum (FBS; Hyclone). The transfection of HEK293T or HeLa cells was carried out with the polyethylenimine method or Lipofectamine 2000 (Invitrogen) according to the manufacturers' instructions.

**Mice.** Lyz-Cre mice were from the Jackson Laboratory; *Atg5*$^{flox/flox}$ mice were a kind gift of N. Mizushima (The University of Tokyo). Both Lyz-Cre mice and *Atg5*$^{flox/flox}$ mice were on C57BL/6 genetic background. *Atg5*$^{flox/flox}$ mice were first crossed with Lyz-Cre mice. The F1 *Atg5* $^{flox/+}$-Lyz-Cre mice then crossed with *Atg5*$^{flox/flox}$ mice to generate *Atg5*$^{flox/flox}$-Lyz-Cre mice as macrophage-specific *Atg5* $^{–/–}$ mice and *Atg5*$^{flox/flox}$ mice as controls. All mice were housed in a specific pathogen-free (SPF) facility on the basis of standard humane animal husbandry protocols, which were approved by the animal care and use committee of the Institute of Microbiology (Chinese Academy of Sciences).

**Preparation of bone marrow-derived macrophages.** Bone marrow-derived macrophages (BMDMs) were collected from tibiae and femurs of 8–12-weeks old mice. After lysis of red blood cells, BMDMs were cultured in DMEM supplemented with 10% FBS, 1% Penicillin-Streptomycin Solution (Caisson) and Murine M-CSF (Pepro Tech) for 4–6 days.

**Generation of CRISPR/Cas9 knockout cell lines.** The 20-bp *Atg5* or *p62*-tar-geting sequences (see Supplementary Data 2) were predicted by the CRISPR design tool (http://crispr.mit.edu/) and recombined into pSpCas9(BB)-2A-GFP (Addgene). The recombinant pSpCas9(BB)-2A-GFP plasmid was transfected into $4 \times 10^6$ RAW264.7 cells for 24 h. GFP-positive cells were sorted into single clones in 96-well plates by flow cytometry using the BD FACSAria III cell sorter (BD Biosciences). Single clones were identified by immunoblot analysis and PCR sequencing after 2 weeks of culture.

**Mouse infection.** All experiments were performed with age- and sex-matched groups of 8–12-weeks old mice. *Atg5*$^{flox/flox}$-Lyz-Cre mice or *Atg5*$^{flox/flox}$ mice were infected with $1.0 \times 10^5$ CFU of Mtb H37Rv strains by intratracheal injection as previously described[11]. Lung tissues were used for quantitative PCR (qPCR), or homogenized and plated on 7H10 agar plates for CFU counting, or subjected to section and stain with hematoxylin and eosin, or Ziehl-Neelsen acid-fast stain. All animal studies were approved by the Biomedical Research Ethics Committee of Institute of Microbiology (Chinese Academy of Sciences).

**Macrophage infection and colony-forming unit counting.** RAW264.7 cells or BMDMs were seeded in 6-well plates at a density of $5.0 \times 10^5$ cells/well and pre-cultured for 12 h before infection. Frozen mycobacterial strains were thawed at 37 °C and centrifuged. The bacterial pellet was then thoroughly resuspended in DMEM medium with 0.05% Tween-80 by vortexing. Thereafter, cells were infected with Mtb strains at a multiplicity of infection (MOI) of 1 or 5 as indicated in the legends. The media were discarded after incubation with Mtb for 1 h. The cells were washed with PBS buffer for three times and were then incubated again with the fresh DMEM medium. For bacterial colony-forming unit (CFU) counting, cells were washed thrice with PBS buffer and lysed in 7H9 broth with 0.05% SDS at each time point. Several sets of serially gradient dilution of the lysates were prepared in 7H9 broth and then cultivated on 7H10 agar plates. The colonies were counted after 3–4 weeks.

**Protein purification.** *E. coli* BL21 (DE3) strain was used for bacterial expression of GST-tagged and His$_6$-tagged WT or truncated Mtb Rv1468c and Ub (which was cloned into pGEX-6P-1 and pET30a vector). The bacterial strains were grown in LB medium at 37 °C until the OD600 = 0.6. Isopropyl-β-D-thiogalactopyranoside (IPTG) was subsequently added to a final concentration of 250 μM and cultures were shaking for further growing at 16 °C for 16 h. Cells were then harvested by centrifugation at 6500×g for 10 min and suspended in a buffer containing 20 mM Tris-HCl (pH 7.5) and 150 mM NaCl for cell-breaking with using Low-temperature Ultra-high Pressure Continuous Flow Cell Disrupters (JNBIO) or Ultrasonic Homogenizer (Scientz Biotechnology). Glutathione Sepharose 4B (GE Healthcare) and Ni-NTA Agarose (Qiagen) were respectively used for purification of GST-tagged proteins or His$_6$-tagged proteins by means of affinity chromatography, followed by size exclusion chromatography on a Superose 6 Increase 10/300 GL column (GE Healthcare). A total of 5 mg His$_6$-tagged Rv1468c were then used for preparation and purification of anti-Rv1468c antibody by GenScript Biotechnology.

**qPCR analysis.** Lung tissues of mice were used for total RNA extraction with RNeasy Plus Mini Kit (Qiagen). For Mtb-infected macrophages, approximately $5 \times 10^6$ cells were harvested and lysed. The total RNA from mycobacteria was then extracted with a Bacterial RNA Extraction Kit (Dongsheng Biotech). The reverse-transcription of RNA was accomplished by using a 1st Strand cDNA Synthesis SuperMix (Yeasen) and performed to qPCR analysis with KAPA SYBR FAST qPCR Kit (KAPA Biosystems) on ABI 7300 system (Applied Biosystems) as previously described[11]. Quantitative expression of targeted gene was normalized to *Gapdh* for mouse genes or 16S ribosomal RNA for mycobacterial genes. All qPCR primers were listed in Supplementary Data 2.

**Cell staining and immunofluorescence microscopy.** Alexa Fluor 488 succinimidyl ester (Invitrogen) was used for bacteria staining. Briefly, the bacterial strains were washed thrice with Hanks' Balanced Salt Solution (Beyotime) containing 0.05% Tween-80 by vortexing and resuspended in the buffer with adequate dye for 30 min at 37 °C. Then the bacterial was again extensively washed and prepared in DMEM medium with 0.05% Tween-80 for infection. HeLa cells or macrophage cells were seeded on poly-lysine-coated coverslips and transfected or infected as described above. At designated time points, cells were gently washed with PBS buffer and fixed in 4% paraformaldehyde (PFA) for 15 min at RT. The cells were washed again and then permeabilized with 0.5% Triton X-100 for 5 min. After three washes, the cells were blocked with 2% BSA for 30 min, subsequently incubated in indicated primary antibodies diluted in 2% BSA for 1 h. The cells then washed again, and incubated in Alexa Fluor 594 secondary antibodies (Molecular Probes) for another hour. After successively washed with PBS buffer and deionized water, the coverslips were mounted onto glass slides using DAPI Staining Solution (Leagene Biotechnology). Confocal images were taken with Olympus FV1000 confocal microscope and analyzed by FV10-ASW 3.0 software.

**Immunoblot analysis and immunoprecipitation.** Cells were lysed in the Cell Lysis Buffer for Western and IP (Beyotime) or RIPA Lysis Buffer (Beyotime) supplemented with 1 mM phenylmethanesulfonyl fluoride (PMSF). Proteins were separated by SDS-PAGE and transferred to polyvinylidene difluoride membranes (Millipore). The membranes were blocked with 5% skimmed milk powder in Tris-buffered saline with 0.1% Tween-20 (TBST) for 1 h at room temperature (RT) and subsequently incubated with primary antibodies overnight at 4 °C. The membranes were then incubated with goat anti-mouse IgG or goat anti-rabbit IgG conjugated to HRP for 1 h at RT after three washes of 10 min each with TBST. Finally, the membranes were developed by Immobilon Western Chemiluminescent HRP Substrate (Millipore) after three washes with TBST again and exposed to X-ray film. For immunoprecipitation, cells were lysed in a lysis buffer containing 50 mM Tris-HCl (pH 7.4), 150 mM NaCl, 1 mM EDTA, 1% Triton X-100 and 1% protease inhibitor cocktail (Bimake). Cell lysates were incubated with anti-Flag M2 Affinity Gel (Sigma-Aldrich) for immunoprecipitation of Flag-tagged in a vertical rotator (Scientz Biotechnology) at 4 °C for 5 h. After five washes with lysis buffer, the immunocomplexes bound in affinity beads were analyzed by SDS-PAGE and blotted with indicated antibodies.

**Cell fractionation and proteinase K sensitivity assay.** Mycobacterial strains were grown in 50 ml of 7H9 medium until mid-log phase and harvested by centrifugation at 3000×g for 15 min at 4 °C. Each sample of the subcellular fraction was isolated by ultracentrifugation[18]. The bacterial pellets were washed twice with PBS buffer and resuspended in 10 ml of PBS buffer added with 1% protease inhibitor cocktail. Cells were then sonicated by Ultrasonic Homogenizer with an operating cycle of 3 s on and 7 s off at 200 W for 15 min on ice. Fragmented cells were centrifuged at 3000×g, 4 °C for 15 min to precipitate cell debris. The supernatants were transferred into fresh tubes and centrifuged at 27,000×g at 4 °C for 30 min in an Optima XE-100 ultracentrifuge (SW41Ti rotor) to precipitate cell wall. The supernatants were collected and precipitated with 10% trichloroacetic acid (TCA) and washed with 80% acetone for preparation of bacterial cytosolic proteins. For proteinase K sensitivity assay, 20 ml logarithmic phase bacterial cultures were harvested, subsequently washed and resuspended in PBS buffer. Cells were then equally divided into two aliquots, one of which was incubated with 100 μg/ml proteinase K (Merck) in a vertical rotator at 4 °C for 15 min. Protease inhibitor cocktail was added into samples to terminate the reaction followed by two washes with PBS buffer.

**In vitro Mtb-Ub binding assay.** A total of 8 ml logarithmic phase bacterial cultures were harvested, subsequently washed and resuspended in PBS buffer. Cells were then equally divided into two aliquots, one of which was subjected to short exposure with proteinase K as described before. After extensive wash with PBST buffer, bacteria were resuspended and incubated with 10 μg of K63-linked poly-Ub (Ub$^{2-7}$) or K48-linked poly-Ub (Ub$^{2-7}$) in 500 μl PBST buffer supplemented with 0.1 mg/ml BSA. After incubation at 4 °C for 4 h, bacteria were extensively washed with PBST buffer and were precipitated by centrifugation at 3600×g at 4 °C for 5 min. For immunoblot analysis, bacteria were resuspended in an appropriate volume of 1 × SDS-PAGE loading buffer followed by boil for 10 min to obtain the samples of bacterial lysates. For immunofluorescence assay, bacteria were resuspended in 4% paraformaldehyde (PFA) for 15 min at RT for fixation followed by blocking in PBST containing 2% BSA for 30 min at RT, and immunostained by using anti-Ub antibody as described before to prepare the samples for detection.

**In vitro precipitation assay.** For the precipitation assay, 10 μg of GST and GST-protein fusions GST-Rv1468c and indicated truncated proteins were immobilized onto 20 μl of Glutathione Sepharose 4B resins in 500 μl of binding buffer (50 mM Tris, pH 7.5, 150 mM NaCl, 5 mM DTT and 0.1% NP-40) supplemented with 1% protease inhibitor cocktail for 1 h at 4 °C. The resins were then washed three times with binding buffer and incubated with 10 μg of purified His-tagged mono-Ub, K63-linked poly-Ub (Ub$^{2-7}$) or K48-linked poly-Ub (Ub$^{2-7}$) in 500 μl binding buffer supplemented with 0.1 mg/ml BSA. After 4 h of incubation at 4 °C, beads were extensively washed and the bound protein complexes were subjected to SDS-PAGE and blotted with indicated antibodies.

**Selective permeabilization with digitonin.** Macrophage cells seeded on coverslips were infected with dyed Mtb strains as described before. Cells were washed thrice with KHM buffer (110 mM KOAc, 20 mM HEPES, 2 mM MgCl$_2$, pH 7.2) and treated by 50 μg/ml digitonin (Abcam) diluted in KHM buffer at RT for 1 min to selectively permeabilize the plasma membranes[17]. After immediately washed with KHM buffer, coverslips were submerged in blocking buffer (PBS buffer containing 1% BSA) for 10 min and then incubated with indicated primary antibodies at 30 °C for 20 min. After washed thrice with PBS buffer, coverslips were incubated in Alexa Fluor 594 secondary antibodies (Molecular Probes). For complete permeabilization of cell membrane as control, cells were further treated with 0.2% Triton X-100 after digitonin permeabilization, followed by blocking and incubation with antibodies as described above. Stained cells were washed extensively and subjected to immunofluorescence analysis.

**Transmission electron microscopy.** BMDMs infected with mycobacteria were collected by centrifugation and washed by PBS buffer for three times and fixed with 2.5% glutaraldehyde overnight. Cells were post-fixed with 1% osmium tetroxide for 2 h, dehydrated in a graded series of ethanol and embedded in SPI-PON 812 resin and polymerized for 24 h at 60 °C. Ultrathin sections (~70 nm) were cut by microtome (Leica EM UC7), mounted on copper grids, and stained with uranyl acetate and lead citrate. For immunoelectron microscopy, ultrathin cryosections of mycobacterial cells or BMDMs were prepared[69]. Briefly, the cells were fixed with 4% PFA and 0.5% glutaraldehyde in PBS at 4 °C for 2 h, and were subsequently embedded with 12% gelatin and infiltrated with 2.3 M sucrose in PBS overnight at 4 °C, and were then frozen in liquid nitrogen and cryosectioned (Tokuyasu method). Ultrathin cryosections were obtained by Leica EM FC7 microtome, which were then collected on grids using sucrose and methylcellulose. The rabbit anti-Rv1468c antibody (1:200 diluted), mouse anti-Ub (1:10 diluted), or rabbit anti-LC3B (1:50 diluted) were used for detection of designated proteins on the grids containing mycobacterial cells or BMDMs infected with mycobacteria. The goat anti-rabbit 10-nm or 6-nm gold conjugates (1:200 diluted; Aurion), or the goat anti-mouse 10-nm gold conjugates (1:200 diluted; Aurion) were used for immunolabelling. The grids were incubated for 1 h with primer antibodies and subsequently with second antibodies at room temperature. Finally, samples were stained

with a 2% methylcellulose and 0.4% uranyl acetate solution. Images were taken by FEI Tecnai Spirit 120 kV transmission electron microscope.

**Statistical analysis**. Unpaired two-tailed Student's *t*-tests, one-way ANOVA, or two-way ANOVA analysis followed by multiple comparisons was used for statistical analysis as indicated in the corresponding figure legends. The quantified data with statistical analysis were performed using GraphPad Prism 8.0. $P < 0.05$ was considered to be statistically significant.

**Reporting summary**. Further information on research design is available in the Nature Research Reporting Summary linked to this article.

## Data availability
The source data underlying Figs. 1c, 3c, e, 4a, b, e, f, 5b, d, 6b, 7a, d, e, and 8a, and Supplementary Figs. 1c, 3e, f, 4e, f, 5a, d, e, 6b, c, 7b, d, 8b, 10c, e, g, 11d, and 12 are provided in Source Data. The original data of unprocessed blot and gel images are also available in Source Data. All other data that support the findings of this study are available from the corresponding author upon reasonable request.

## Code availability
The open sources including Jalview 2.10.2b2 (http://www.jalview.org/) and iTols online tool (http://itol.embl.de/) were used for Supplementary Fig. 2a and 2d according to the instruction on their corresponding websites, respectively. The source code and dataset used for Supplementary Fig. 2e with R version 3.4.4 (https://mran.microsoft.com/news) are available on GitHub (https://github.com/LuShuYangMing/Protein-Domain-Plot), where an expected output and a brief instruction are attached.

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

## Acknowledgements
This work was supported by the National Key Research and Development Program of China (Grant 2017YFA0505900 to C.H.L.), the Major Program of National Natural Science Foundation of China (Grant 31830003 to C.H.L.), the National Natural Science Funds for Distinguished Young Scholar (Grant 81825014 to C.H.L.), the Strategic Priority Research Program of the Chinese Academy of Sciences (Grant XDB29020000 to C.H.L.), the National Basic Research Program of China (Grant 2014CB74440 to C. H.L.), the National Natural Science Foundation of China (Grant 81571954 to C.H.L.), the National Science and Technology Major Project (Grant No. 2018ZX10101004 to J. W.), and the Youth Innovation Promotion Association of the Chinese Academy of Sciences (to C.H.L. and J. W.). We thank N. Mizushima (The University of Tokyo) for *Atg5^{flox/flox}* mice; W. R. Jacobs (Albert Einstein College of Medicine, Yeshiva University), H. Wang (Xiamen University, Xiamen) and F. Shao (National Institute of Biological Sciences, Beijing) for multiple plasmids; L. Sun (Center for Biological Imaging, Institute of Biophysics, Beijing) for helping with electron microscopy analysis; T. Zhao (Institute of Microbiology, Chinese Academy of Sciences, Beijing) for helping with flow cytometry; Y. Li (Fudan University Shanghai Medical College) and X. Liu (Cincinnati Children's Hospital Medical Center, Cincinnati) for providing experimental suggestions.

## Author contributions
C.H.L. conceived the project; C.H.L. and Q.C. designed the experiments and analyzed the data; Q.C., X.W., L.Q., Y.Z., P.G., Z.L., B.L., J.W., Y.Z.Z. and W.D. performed the experiments; L.Z., W.L., D.Z., Y.P., and G.F.G. contributed some critical experimental materials; Q.C. and C.H.L. wrote the manuscript, with critical input from all other authors; all authors read and approved the final version of the manuscript.

## Additional information

**Competing interests:** The authors declare no competing interests.

