## [Peer Review File · Nature Communications]

Reviewers' comments:

Reviewer #1 (Remarks to the Author):

In the manuscript entitled "Ubiquitin is an innate immune sensor of the mycobacterial surface protein for xenophagy initiation" by Chai et al, the authors describe an Mtb surface protein Rv1468c that binds directly to ubiquitin chains to target Mtb to autophagy to control bacterial replication. The authors demonstrate that deleting Rv1468c decreases targeting, complementing with WT Rv1468c rescues it, but complementing with a Ub binding-deficient Rv1468c mutant fails to. In vivo, mice infected with either of these mutants have elevated bacterial burdens and hyperinflammatory immune responses.

Overall, this manuscript is a thorough and detailed study of a very intriguing new mechanism of how Mtb is tagged with ubiquitin during infection. The data itself is convincing, and many important controls are included to support the authors' conclusions. The bacterial genetics in this study provide especially powerful evidence for this surface protein's role in ubiquitin recruitment.

Major Concerns:

1. In the title and in several places throughout the paper (lines 40, 320-322), the authors describe this phenomenon as a host driven pathway for detecting Mtb (by calling ubiquitin a host "innate immune sensor"). However, Rv1468c is a bacterial protein that Mtb uses to recruit ubiquitin to itself, which makes it a bacterial-driven process. The bacteria may be taking advantage of host biology by recruiting host ubiquitin chains, but it doesn't seem like the host is actively using ubiquitin to sense anything. Therefore, describing this process as host innate immune recognition seems inaccurate.
2. The different degrees of targeting for the WT and mutant strains could be due to differences in their ability to permeabilize the phagosome and access the cytosol. The authors do acknowledge this and address it by staining for galectin-3+ bacteria. We believe that galectin-3 alone is not the best measure of phagosomal membrane integrity and since much of the data hinges on the wt and the mutant Rv1468c having similar (the same) permeabilized phagosomes, that more experiments should be done to rule out this important possibility. The authors could utilize their digitonin permeabilization assay from Supp Fig 1 to measure how many bacteria of each strain have cytosolic access, and/or the authors could use EM as in Fig 5c and d but at earlier time points (4 hr) to determine how many of each strain are in the cytosol vs phagosomes.

3. The authors have provided a good deal of evidence to support the idea that Rv1468c can bind directly to ubiquitin in vitro, and their bacterial mutants provide strong evidence that this process is important in vivo. However, we think that a few more control experiments might help bolster their conclusions even more. First, we'd like to see immunofluorescence data (as in Fig 1c) in Fig 2 to show that when expressed in mycobacteria, WT but not mutant Rv1468c binds to ubiquitin chains. Second, it would be good to see ubiquitin binding by both immunofluorescence and western blots with *M. smegmatis* expressing either WT or mutant Rv1468c – in this reconstituted system, does Rv1468c alone confer UB-binding ability? This would further rule out the possibility that ubiquitin might bind nonspecifically to the outer membrane of mycobacterial species and confirm that the UBA domain is on the outer surface and accessible to host proteins.

Minor Concerns:

1. The data presented in Supp Fig 1 doesn't seem to fit with the rest of the manuscript, and despite the conclusions the authors draw from it, the data doesn't support the idea that ubiquitin is bound directly to the bacteria. Instead of discussing this data, it might benefit the manuscript to describe in more detail the first half of Fig 1, which is strong data that is critical for setting up the remainder of the paper.

2. In the discussion, the authors should speculate on the nature of the ubiquitin chains. Are they attached to substrates? Are they free-floating Ub chains?

3. Also in the discussion, it would be helpful for the authors to more directly address how/why *Mtb* has evolved to express a protein that directly leads to killing in a host. This is alluded to in several ways, but a direct discussion of this relatively counterintuitive concept would be helpful.

4. Why were the experiments in Figure 7a and b done in RAW cells while the rest of the figure and the remainder of the paper was done using BMDMs? These results are difficult to compare to the rest of the results in the paper due to the change in cell type.

5. Did the authors look at ubiquitin accumulation in Atg5^{-/-} macrophages? Showing amplified accumulation around WT but not mutant bacteria would further support their model.

6. In Fig 1 and 2, there are Western blots labeled as bacterial lysate while the results describe the experiments as using intact bacteria. This is confusing without reading the methods to determine

exactly what was done. Editing the labels on these figures would help clarify what is being done/looked at in this experiment.

7. The authors should make an effort to reference primary literature rather than reviews wherever possible. In some places, references were overlooked, such as in line 329, where cGAS is also required for targeting Mtb to selective autophagy (Collins et al., Cell Host Microbe, 2015 and Watson et al., Cell Host Microbe, 2015).

8. The labels on Supp Fig 2 are extremely difficult to read.

9. On line 187, the text reads “1-fold” but should say “2-fold”.

10. On line 250, Rv1468c is referred to as “Ub-modified”, but the model is that it is bound to and not modified by ubiquitin.

Reviewer #2 (Remarks to the Author):

The MS by Chai et al. reports an interesting new finding that is likely to be relevant to the complex host-pathogen relationship in *M. tuberculosis* infection. The basic result is that one Mtb protein, Rv1468c, is found to have a ubiquitin binding domain and binds ubiquitin at the surface of the bacterial cell. A very extensive set of studies is presented showing that Rv1468c, a member of the PE_PGRS family, is expressed on the bacterial surface where it noncovalently binds different forms of polyubiquitin. They also show that this takes place in infected myeloid cells, and that it drives a p62/LC3-dependent xenophagy process that reduces bacterial survival in cell culture and during mouse infection at early time points (2-3 weeks). Overall, it is an excellent study with an impressive array of high quality results to support their conclusions. I have one major comment on the MS for the authors' consideration, and a few relatively minor technical issues.

Major comment: I agree with the authors' conclusion that Rv1468c is a protein on the surface of the bacterium that binds polyubiquitin chains, potentially promoting clearance and destruction of the bacteria by xenophagy. This is consistent with what they are seeing in Fig. 6a, which shows that

expression of Rv1468c leads to modest attenuation of growth in mice with intact xenophagy. What they have not addressed in their MS is why Mtb would retain such a protein and its highly conserved UBA domain if its only function is to mark bacteria for destruction. Unless the UBA domain has some other function, its retention and marked conservation suggests that it must somehow favor the successful propagation or transmission of the bacteria. What might this function be?

Author has discussed little about this in the discussion section and proposed it as a long-term intracellular survival to avoid excessive host inflammatory immune responses. However, this idea seems to be murky as this protein expressed both at early and late phase of infection (Kruh et al., 2010).

They don't discuss this, or provide any insight into how the expression of this protein might do this, or how it obtains an overall advantage to having it. Possibly, Rv1468c is acting as an attenuator of virulence to promote indolent growth of the bacilli in endosomes by eliminating cytosolic bacilli. This might have a net effect of driving prolonged infections with increased pulmonary cavitation and greater transmission. I realize this involves speculation beyond the actual data that are shown, but it would enhance the MS to have some idea of how the authors are thinking about this and how they plan to move the work forward in the future (especially with regard to their stated idea of "a potential target for the development of novel pathogen-host interfaces-based TB treatments").

Point-by-point responses to reviewers' comments

Re: Chai, et al., “Ubiquitin binding to the mycobacterial surface protein as an innate immune trigger to initiate xenophagy” (NCOMMS–18–33181).

Reviewers' comments:

Reviewer #1

Remarks to the author:

In the manuscript entitled “Ubiquitin is an innate immune sensor of the mycobacterial surface protein for xenophagy initiation” by Chai et al, the authors describe an Mtb surface protein Rv1468c that binds directly to ubiquitin chains to target Mtb to autophagy to control bacterial replication. The authors demonstrate that deleting Rv1468c decreases targeting, complementing with WT Rv1468c rescues it, but complementing with a Ub binding-deficient Rv1468c mutant fails to. *In vivo*, mice infected with either of these mutants have elevated bacterial burdens and hyperinflammatory immune responses.

Overall, this manuscript is a thorough and detailed study of a very intriguing new mechanism of how Mtb is tagged with ubiquitin during infection. The data itself is convincing, and many important controls are included to support the authors' conclusions. The bacterial genetics in this study provide especially powerful evidence for this surface protein's role in ubiquitin recruitment.

R: We thank the reviewer for the encouraging comments on our manuscript.

Major Concerns:

1. In the title and in several places throughout the paper (lines 40, 320–322), the authors describe this phenomenon as a host driven pathway for detecting Mtb (by calling ubiquitin a host “innate immune sensor”). However, Rv1468c is a bacterial protein that Mtb uses to recruit ubiquitin to itself, which makes it a bacterial-driven process. The bacteria may be taking advantage of host biology by recruiting host ubiquitin chains, but it doesn't seem like the host is actively using ubiquitin to sense anything. Therefore, describing this process as host innate immune recognition seems inaccurate.

R: We thank the reviewer for raising this concern. Indeed, it is difficult to define whether the host or the pathogen initiates and drives the process of ubiquitin binding to mycobacterial surface protein Rv1468c, which represents as a result of the long-term dynamic coevolution and antagonism between the host and the pathogen. From the host perspective, it seems explainable that the host is able to actively recognize some conserved molecular patterns (such as lipoarabinomannan, and may also include the Rv1468c UBA domain as discovered in this study) on mycobacterial surface to initiate immune defense activation. This notion could be supported by the finding that the large majority

of known human T cell epitopes in Mtb are hyperconserved for recognition by the host^{1,2}. Thus, Rv1468c UBA domain may represent a unique pathogen molecular pattern actively sensed by host using ubiquitin for triggering the antimicrobial autophagy. While from the pathogen's perspective, it is also reasonable to suspect that Rv1468c-mediated ubiquitin recruitment on Mtb surface might be a bacterial-driven process, which could somehow exert certain beneficial advantages to the pathogen.

Although it is hard to define whether the host or the pathogen drives the process of ubiquitin binding to mycobacterial surface protein Rv1468c, it has been clearly demonstrated in our study that ubiquitin binding to Rv1468c is a crucial event to elicit xenophagy, which is an important host innate immune mechanism for pathogen clearance. Nevertheless, as suggested by the reviewer, we should be cautious in describing the ubiquitin as an innate immune sensor, which implying that the ubiquitin attachment to Mtb surface is a host-driven process. Therefore, we adopted a relatively neutral tone and revised the title as "Ubiquitin binding to the mycobacterial surface protein as an innate immune trigger to initiate xenophagy". We also modified the relevant descriptions accordingly in the revised manuscript.

2. The different degrees of targeting for the WT and mutant strains could be due to differences in their ability to permeabilize the phagosome and access the cytosol. The authors do acknowledge this and address it by staining for galectin-3⁺ bacteria. We believe that galectin-3 alone is not the best measure of phagosomal membrane integrity and since much of the data hinges on the wt and the mutant Rv1468c having similar (the same) permeabilized phagosomes, and that more experiments should be done to rule out this important possibility. The authors could utilize their digitonin permeabilization assay from Supp Fig 1 to measure how many bacteria of each strain have cytosolic access, and/or the authors could use EM as in Fig 5c and d but at earlier time points (4 hr) to determine how many of each strain are in the cytosol vs phagosomes.

R: We thank the reviewer for this constructive suggestion. We agree with the reviewer that additional experiments should be done to rule out the possibility that Rv1468c might be involved in phagosomal damage. We thus performed both digitonin permeabilization and electron microscopy analysis as suggested by the reviewer to further confirm our finding. As shown in revised Supplementary Fig. 7, the Mtb Rv1468c variant strains exhibited similar accessibility to anti-Mtb antibody when macrophage membranes were selectively permeabilized by digitonin, and showed no significant difference in the proportion of bacteria in the cytosol and in the phagosomes at an early time point (4 hours) of infection in macrophages. Thus, these results, together with the confocal microscopy data from experiment using galectin-3 (revised Supplementary Fig. 8), help rule out the possibility that the different degrees of targeting for the WT and mutant Mtb strains could be due to differences in their ability to permeabilize the phagosome

for cytosol accession.

3. The authors have provided a good deal of evidence to support the idea that Rv1468c can bind directly to ubiquitin *in vitro*, and their bacterial mutants provide strong evidence that this process is important *in vivo*. However, we think that a few more control experiments might help bolster their conclusions even more. First, we'd like to see immunofluorescence data (as in Fig 1c) in Fig 2 to show that when expressed in mycobacteria, WT but not mutant Rv1468c binds to ubiquitin chains. Second, it would be good to see ubiquitin binding by both immunofluorescence and western blots with *M. smegmatis* expressing either WT or mutant Rv1468c – in this reconstituted system, does Rv1468c alone confer UB-binding ability? This would further rule out the possibility that ubiquitin might bind nonspecifically to the outer membrane of mycobacterial species and confirm that the UBA domain is on the outer surface and accessible to host proteins.

R: We thank the reviewer for pointing out this issue and for the constructive suggestion. We have performed the immunofluorescence assay to confirm that deletion of mutation of Rv1468c in Mtb dramatically reduced its ability to bind to the ubiquitin chains *in vitro* (revised Supplementary Fig. 3c–f). Furthermore, by using both immunoblot analysis and immunofluorescence assays, we observed that *M. smegmatis* expressing WT Rv1468c, but not WT *M. smegmatis* and *M. smegmatis* expressing Rv1468c L65G, had enhanced interaction with either K63 or K48 poly-Ub chains *in vitro* (revised Supplementary Fig. 4). Therefore, these results further confirmed that ubiquitin could directly bind to Mtb Rv1468c UBA domain, rather than bind nonspecifically to the outer membrane of mycobacterial species.

Minor Concerns:

1. The data presented in Supp Fig 1 doesn't seem to fit with the rest of the manuscript, and despite the conclusions the authors draw from it, the data doesn't support the idea that ubiquitin is bound directly to the bacteria. Instead of discussing this data, it might benefit the manuscript to describe in more detail the first half of Fig 1, which is strong data that is critical for setting up the remainder of the paper.

R: We thank the reviewer for raising this issue. We have revised the manuscript for more detailed description of Figure 1a–c (please see the first paragraph of the Results section). Also, we humbly suggest that the data in Supplementary Fig. 1 are not redundant based on the following explanations: Initially, we observed that a certain number of Mtb did present in the cytosol after infection for 4 hours by using electron microscope (Supplementary Fig. 1a), which corroborated the previous findings that Mtb could damage and escape from the phagosomes at a very early time of infection³⁻⁵, making it logical to hypothesize that host cytosolic ubiquitin could access the mycobacterial surface for direct interaction. Besides, by immunofluorescence assay with different permeabilization methods, we verified that during Mtb infection in macrophages, a considerable proportion

of ubiquitin directly adhered to their surface (Supplementary Fig. 1b, c). Therefore, these *in vivo* experiments helped support the conclusion that ubiquitin could directly bind to Mtb surface during the infection, which together with the *in vitro* data shown in Figure 1a–c, serve as a necessary prerequisite for us to further test if and which protein on Mtb surface could mediate the binding of ubiquitin to bacteria.

2. In the discussion, the authors should speculate on the nature of the ubiquitin chains. Are they attached to substrates? Are they free-floating Ub chains?

R: We thank the reviewer for pointing this out. We added our speculation on the nature of the ubiquitin chains in the revised manuscript (please see the second paragraph in the Discussion section). Based on our data from *in vitro* binding assays, Mtb Rv1468c is able to bind to free-floating poly-Ub chains, which Ub chains in eukaryotic cells are emerging as key factors at the interface of the host-pathogen interactions⁶⁻⁸. Nevertheless, since both free-floating and substrate-attached poly-Ub chains exist within host cells, we do not exclude the possibility that a portion of Rv1468c could also interact with certain substrate-attached poly-Ub chains during Mtb infection in macrophages, and the potential regulatory function of those interactions warrants future investigation.

3. Also in the discussion, it would be helpful for the authors to more directly address how/why Mtb has evolved to express a protein that directly leads to killing in a host. This is alluded to in several ways, but a direct discussion of this relatively counterintuitive concept would be helpful.

R: We thank the reviewer for this kind suggestion. We have provided a more detailed perspective on why Mtb would retain such as a protein like Rv1468c that directly leads to killing in a host (please see the third paragraph in the Discussion section).

4. Why were the experiments in Figure 7a and b done in RAW cells while the rest of the figure and the remainder of the paper was done using BMDMs? These results are difficult to compare to the rest of the results in the paper due to the change in cell type.

R: We thank the reviewer for pointing out this issue. We are sorry for causing this confusion. Actually, all of the data from RAW264.7 cells-based experiments in this study were presented in previous Figure 7a and previous Supplementary Fig. 7 as described in previous version of the manuscript.

To make ourselves clearer, we have re-organized our data and revised the manuscript accordingly. Basically, we used two complementary experimental systems (include generating Atg5-deficient RAW264.7 macrophage cell line by using the CRISPR/Cas9 system⁹, and obtaining *Atg5*^{-/-} BMDMs from *Atg5*^{flx/flx}-*Lyz-Cre* KO mice) to confirm that ubiquitin-Rv1468c

interaction-triggered anti-Mtb process is mediated by host selective autophagy pathway. Initially, we deleted p62 or Atg5 in RAW264.7 macrophages using the CRISPR/Cas9 system to preliminary confirm that disruption of host selective autophagy pathway could impair the Rv1468c-ubiquitin interaction-triggered antimycobacterial process (revised Supplementary Fig. 10). Encouraged by the results from *Atg5*^{-/-} RAW264.7 macrophages, we then further obtained the *Atg5*^{flox/flox} mice and *Lyz-Cre* mice to generate *Atg5*^{flox/flox}-*Lyz-Cre* mice, from which we can directly get *Atg5*^{-/-} BMDMs for further confirmation of our finding. Using *Atg5*^{-/-} BMDMs, we observed consistent results as that from *Atg5*^{-/-} RAW264.7 cells (revised Figure 7). Thus, our data obtained from RAW264.7 cells and BMDMs provide consistent evidence supporting our conclusion that subversion of host selective autophagy pathway by deletion of Atg5 impairs the Rv1468c-ubiquitin binding-dependent antimycobacterial process.

5. Did the authors look at ubiquitin accumulation in *Atg5*^{-/-} macrophages? Showing amplified accumulation around WT but not mutant bacteria would further support their model.

R: We thank the reviewer for raising this issue. We performed immunofluorescence assay and found that as compared to the control macrophages, the colocalizations of Mtb expressing WT Rv1468c, but not Mtb Δ *Rv1468c* or Mtb Δ *Rv1468c*:*Rv1468c* L65G, with ubiquitin were increased in *Atg5*^{-/-} macrophages (revised Supplementary Fig. 11c, d). This phenomenon could be explained by the possibility that host autophagy flux for degradation of ubiquitin-bound bacteria in *Atg5*^{-/-} macrophages was blocked, so that there were increased numbers of ubiquitin-attached WT Mtb and Mtb Δ *Rv1468c*:*Rv1468c* (but not the Mtb Δ *Rv1468c* and Mtb Δ *Rv1468c*:*Rv1468c* L65G strains that could not efficiently bind to ubiquitin) persisting in the cytosol. This result further confirmed that Rv1468c UBA domain is required for Ub accumulation around Mtb in macrophages.

6. In Fig 1 and 2, there are Western blots labeled as bacterial lysate while the results describe the experiments as using intact bacteria. This is confusing without reading the methods to determine exactly what was done. Editing the labels on these figures would help clarify what is being done/looked at in this experiment.

R: We thank the reviewer for pointing out this issue and for the constructive suggestion. We are sorry for causing this confusion. We modified the relevant places in the Results section, the Methods section, and the legend section for Fig. 1 and 2 in the revised manuscript to make ourselves clearer.

7. The authors should make an effort to reference primary literature rather than reviews wherever possible. In some places, references were overlooked, such as in line 329, where cGAS is also required for targeting Mtb to selective autophagy

(Collins et al., Cell Host Microbe, 2015 and Watson et al., Cell Host Microbe, 2015).

R: We thank the reviewer for pointing out this issue. We added those overlooked references as suggested by the reviewer and cited the primary literatures instead of reviews whenever possible.

8. The labels on Supp Fig 2 are extremely difficult to read.

R: We thank the reviewer for pointing out this issue. We rearranged the images in Supplementary Fig. 2 with enlarged labels to make it clearer.

9. On line 187, the text reads “1-fold” but should say “2-fold”.

R: We thank the reviewer for pointing this out. We have corrected this mistake in the revised manuscript.

10. On line 250, Rv1468c is referred to as “Ub-modified”, but the model is that it is bound to and not modified by ubiquitin.

R: We thank the reviewer for pointing this out. We have changed the description of “Ub-modified” to “Ub-bound” in the revised manuscript.

Reviewer #2

Remarks to the author:

The MS by Chai et al. reports an interesting new finding that is likely to be relevant to the complex host-pathogen relationship in *M. tuberculosis* infection. The basic result is that one *Mtb* protein, Rv1468c, is found to have a ubiquitin binding domain and binds ubiquitin at the surface of the bacterial cell. A very extensive set of studies is presented showing that Rv1468c, a member of the PE_PGRS family, is expressed on the bacterial surface where it noncovalently binds different forms of polyubiquitin. They also show that this takes place in infected myeloid cells, and that it drives a p62/LC3-dependent xenophagy process that reduces bacterial survival in cell culture and during mouse infection at early time points (2-3 weeks). Overall, it is an excellent study with an impressive array of high quality results to support their conclusions. I have one major comment on the MS for the authors' consideration, and a few relatively minor technical issues.

R: We thank the reviewer for the encouraging comments on our manuscript.

Major comment:

I agree with the authors' conclusion that Rv1468c is a protein on the surface of the bacterium that binds polyubiquitin chains, potentially promoting clearance and destruction of the bacteria by xenophagy. This is consistent with what they are seeing in Fig. 6a, which shows that expression of Rv1468c leads to modest attenuation of growth in mice with intact xenophagy. What they have not addressed in their MS is

why Mtb would retain such a protein and its highly conserved UBA domain if its only function is to mark bacteria for destruction. Unless the UBA domain has some other function, its retention and marked conservation suggests that it must somehow favor the successful propagation or transmission of the bacteria. What might this function be?

Author has discussed little about this in the discussion section and proposed it as a long-term intracellular survival to avoid excessive host inflammatory immune responses. However, this idea seems to be murky as this protein expressed both at early and late phase of infection (Kruh et al., 2010).

They don't discuss this, or provide any insight into how the expression of this protein might do this, or how it obtains an overall advantage to having it. Possibly, Rv1468c is acting as an attenuator of virulence to promote indolent growth of the bacilli in endosomes by eliminating cytosolic bacilli. This might have a net effect of driving prolonged infections with increased pulmonary cavitation and greater transmission. I realize this involves speculation beyond the actual data that are shown, but it would enhance the MS to have some idea of how the authors are thinking about this and how they plan to move the work forward in the future (especially with regard to their stated idea of "a potential target for the development of novel pathogen-host interfaces-based TB treatments").

R: We thank the reviewer for the insightful comments and constructive suggestions on our manuscript. We further performed quantitative PCR analysis to examine the expression of several pro-inflammatory cytokines in the lungs of the infected mice. As shown in Supplementary Fig. 12, we detected increased mRNA levels of pro-inflammatory cytokines including *Tnf*, *Il1b* and *Il6* in the lungs of *Atg5^{fl/fl}* mice infected with Mtb $\Delta Rv1468c$ or Mtb $\Delta Rv1468c:Rv1468c$ L65G as compared to that infected with WT Mtb or Mtb $\Delta Rv1468c:Rv1468c$ at 3 weeks post-infection, which was consistent with increased bacterial loads and increased pathology in the lungs of *Atg5^{fl/fl}* mice infected with Rv1468c-deleted or L65G-mutated Mtb strains.

We have also presented a more detailed discussion on why Mtb would evolutionarily retain Rv1468c with a conserved UBA domain as follows: Initially, it seems counterintuitive that Mtb would retain such a conserved UBA domain-containing protein like Rv1468c that functions to mark bacteria for host clearance and thus avoiding excessive host inflammation. In fact, the long-term intimate interplays between Mtb and the host are quite nuanced, rather than irreconcilably in conflict until one is defeated¹⁰. In about 90% of the infected individuals, Mtb could persist in the host by establishing a latent state instead of causing host severe inflammatory responses and tissue damage, as many severe and acute infection-causing pathogens normally do¹¹. Consistently, current evidence indicate that Mtb strains with relatively attenuated virulence may be better tolerated by patients for long-term, which may ultimately lead to an extended duration of illness and increased Mtb transmission potential^{1,12}. There is also evidence showing that the slowly-replicating phenotypes of Mtb could help

maintain the bacterial population in the host by continuously adapting to dynamic microenvironments in granulomas for prolonged infection¹³. Furthermore, another study suggests that lower induction of pro-inflammatory cytokines in infected individuals might be a contributing factor to the evolutionary success of modern Beijing strains of Mtb, as compared to the ancient strains¹⁴. Thus, Rv1468c UBA domain-triggered autophagy could be a viable evolutionary strategy adopted by Mtb to maintain long-term intracellular survival through self-controlling its intracellular bacterial loads to avoid excessive host inflammatory immune responses, which might favor prolonged infection and greater transmission. Interestingly, proteomic profiling of Mtb revealed that Rv1468c is expressed both at the early and chronic stages of infection¹⁵, and the expression of Rv1468c could be regulated by IS6110 in Mtb¹⁶, further suggesting that the expression and retention of this protein might somehow exert certain beneficial advantages to the pathogen. Probably, Rv1468c-triggered autophagy might help preserve indolent growth of Mtb in vacuolar compartments, since the establishment of a sheltered niche within vacuoles mimicking normal cellular compartments could effectively elude host immune surveillance and pathogen clearance¹⁷. This notion could be supported by a variety of studies demonstrating that the ability to create and maintain a specialized vacuolar organelle that supports bacterial replication is an important survival strategy for many intracellular pathogens such as Mtb, *Legionella pneumophila* and *Brucella abortus*¹⁸⁻²⁰. On the other side, the timely removal of cytosolic bacteria via Rv1468c-triggered autophagy could probably minimize the cytosolic exposure of microbial patterns and balance the intracellular bacterial burden to avoid host excessive inflammatory responses^{21,22}. Nevertheless, the overall effect of the Rv1468c-triggered autophagy on the ultimate fate of intracellular Mtb may depend upon multiple factors, including whether the xenophagy flux, the complete process of xenophagy in which pathogens contained within autophagosomes are digested by lysosomes, is compromised or not in Mtb-infected macrophages²³. Thus, the identification of a eukaryotic-type domain such as UBA in a Mtb surface protein for host immune recognition strengthened the notion that the intimate host-pathogen interactions drive their dynamic coevolution and antagonism, which regulate the diverse outcomes of pathogen persistence and host resistance¹ (please see the third paragraph in the Discussion section).

Therefore, caution need be taken and both benefits and potential unfavorable effects should be considered while choosing to target Rv1468c for TB treatment, since there is a possibility that Rv1468c might be manipulated by the pathogen under certain circumstances to its own advantage (please see the fourth paragraph in the Discussion section).

Once again, we greatly appreciate the reviewers for having helped us improve this manuscript tremendously.

References

- 1 Gagneux, S. Ecology and evolution of *Mycobacterium tuberculosis*. *Nat Rev Microbiol* **16**, 202-213, doi:10.1038/nrmicro.2018.8 (2018).
- 2 Copin, R. *et al.* Sequence diversity in the pe_pgrs genes of *Mycobacterium tuberculosis* is independent of human T cell recognition. *MBio* **5**, e00960-00913, doi:10.1128/mBio.00960-13 (2014).
- 3 van der Wel, N. *et al.* *M. tuberculosis* and *M. leprae* translocate from the phagolysosome to the cytosol in myeloid cells. *Cell* **129**, 1287-1298, doi:10.1016/j.cell.2007.05.059 (2007).
- 4 Houben, D. *et al.* ESX-1-mediated translocation to the cytosol controls virulence of mycobacteria. *Cell Microbiol* **14**, 1287-1298, doi:10.1111/j.1462-5822.2012.01799.x (2012).
- 5 Simeone, R. *et al.* Cytosolic access of *Mycobacterium tuberculosis*: critical impact of phagosomal acidification control and demonstration of occurrence *in vivo*. *PLoS Pathog* **11**, e1004650, doi:10.1371/journal.ppat.1004650 (2015).
- 6 Banerjee, I. *et al.* Influenza A virus uses the aggresome processing machinery for host cell entry. *Science* **346**, 473-477, doi:10.1126/science.1257037 (2014).
- 7 Zeng, W. *et al.* Reconstitution of the RIG-I pathway reveals a signaling role of unanchored polyubiquitin chains in innate immunity. *Cell* **141**, 315-330, doi:10.1016/j.cell.2010.03.029 (2010).
- 8 Rajsbaum, R. *et al.* Unanchored K48-linked polyubiquitin synthesized by the E3-ubiquitin ligase TRIM6 stimulates the interferon-IKKepsilon kinase-mediated antiviral response. *Immunity* **40**, 880-895, doi:10.1016/j.immuni.2014.04.018 (2014).
- 9 Ran, F. A. *et al.* Genome engineering using the CRISPR-Cas9 system. *Nat Protoc* **8**, 2281-2308, doi:10.1038/nprot.2013.143 (2013).
- 10 Chai, Q., Zhang, Y. & Liu, C. H. *Mycobacterium tuberculosis*: an adaptable pathogen associated with multiple human diseases. *Front Cell Infect Microbiol* **8**, 158, doi:10.3389/fcimb.2018.00158 (2018).
- 11 Cambier, C. J., Falkow, S. & Ramakrishnan, L. Host evasion and exploitation schemes of *Mycobacterium tuberculosis*. *Cell* **159**, 1497-1509, doi:10.1016/j.cell.2014.11.024 (2014).
- 12 Dye, C. & Williams, B. G. The population dynamics and control of tuberculosis. *Science* **328**, 856-861, doi:10.1126/science.1185449 (2010).
- 13 Pienaar, E., Matern, W. M., Linderman, J. J., Bader, J. S. & Kirschner, D. E. Multiscale model of *Mycobacterium tuberculosis* infection maps metabolite and gene perturbations to granuloma sterilization predictions. *Infect Immun* **84**, 1650-1669, doi:10.1128/IAI.01438-15 (2016).
- 14 van Laarhoven, A. *et al.* Low induction of proinflammatory cytokines parallels

evolutionary success of modern strains within the *Mycobacterium tuberculosis* Beijing genotype. *Infect Immun* **81**, 3750-3756, doi:10.1128/IAI.00282-13 (2013).

- 15 Kruh, N. A., Troudt, J., Izzo, A., Prenni, J. & Dobos, K. M. Portrait of a pathogen: the *Mycobacterium tuberculosis* proteome *in vivo*. *PLoS One* **5**, e13938, doi:10.1371/journal.pone.0013938 (2010).
- 16 Safi, H. *et al.* IS6110 functions as a mobile, monocyte-activated promoter in *Mycobacterium tuberculosis*. *Mol Microbiol* **52**, 999-1012, doi:10.1111/j.1365-2958.2004.04037.x (2004).
- 17 Kumar, Y. & Valdivia, R. H. Leading a sheltered life: intracellular pathogens and maintenance of vacuolar compartments. *Cell Host Microbe* **5**, 593-601, doi:10.1016/j.chom.2009.05.014 (2009).
- 18 Philips, J. A. Mycobacterial manipulation of vacuolar sorting. *Cell Microbiol* **10**, 2408-2415, doi:10.1111/j.1462-5822.2008.01239.x (2008).
- 19 Barlocher, K., Welin, A. & Hilbi, H. Formation of the *Legionella* replicative compartment at the crossroads of retrograde trafficking. *Frontiers in cellular and infection microbiology* **7**, 482, doi:10.3389/fcimb.2017.00482 (2017).
- 20 Miller, C. N. *et al.* A *Brucella* type IV effector targets the cog tethering complex to remodel host secretory traffic and promote intracellular replication. *Cell host & microbe* **22**, 317-329 e317, doi:10.1016/j.chom.2017.07.017 (2017).
- 21 Mitchell, G. & Isberg, R. R. Innate immunity to intracellular pathogens: balancing microbial elimination and inflammation. *Cell Host Microbe* **22**, 166-175, doi:10.1016/j.chom.2017.07.005 (2017).
- 22 Deretic, V. & Levine, B. Autophagy balances inflammation in innate immunity. *Autophagy* **14**, 243-251, doi:10.1080/15548627.2017.1402992 (2018).
- 23 Chandra, P. & Kumar, D. Selective autophagy gets more selective: uncoupling of autophagy flux and xenophagy flux in *Mycobacterium tuberculosis*-infected macrophages. *Autophagy* **12**, 608-609, doi:10.1080/15548627.2016.1139263 (2016).

REVIEWERS' COMMENTS:

Reviewer #1 (Remarks to the Author):

We really enjoyed reading and reviewing this manuscript. The experiments are thorough, and the data are very convincing. It is a very interesting study that will likely have a significant impact on how the field thinks about host -Mtb interactions. One suggestion is a slight re-wording of the title to "A Mycobacterium tuberculosis surface protein recruits (or binds to) ubiquitin to trigger host xenophagy".

Reviewer #2 (Remarks to the Author):

The authors were incredibly thorough regarding each point of major and minor concern presented by the reviewers. The concerns were adequately addressed and sufficient detail was provided where there was difference of opinion.

Point-by-point responses to reviewers' comments

Re: Chai, et al., "A *Mycobacterium tuberculosis* surface protein recruits ubiquitin to trigger host xenophagy" (NCOMMS-18-33181B).

Reviewers' comments:

Reviewer #1 (Remarks to the Author):

We really enjoyed reading and reviewing this manuscript. The experiments are thorough, and the data are very convincing. It is a very interesting study that will likely have a significant impact on how the field thinks about host -Mtb interactions. One suggestion is a slight re-wording of the title to "A *Mycobacterium tuberculosis* surface protein recruits (or binds to) ubiquitin to trigger host xenophagy".

R: We greatly appreciate the reviewer for having helped us improve this manuscript tremendously. We have further re-worded the title according to the reviewer's suggestion.

Reviewer #2 (Remarks to the Author):

The authors were incredibly thorough regarding each point of major and minor concern presented by the reviewers. The concerns were adequately addressed and sufficient detail was provided where there was difference of opinion.

R: We thank the reviewer for the encouraging comments on our revised manuscript.

Once again, we thank the reviewers for helpful and constructive comments on our manuscript.